# Half-Heusler alloys as emerging high power density thermoelectric cooling materials

Hangtian Zhu [1,2,3] ✉, Wenjie Li[1,3] ✉, Amin Nozariasbmarz [1], Na Liu[1], Yu Zhang [1], Shashank Priya [1] ✉ & Bed Poudel [1] ✉

To achieve optimal thermoelectric performance, it is crucial to manipulate the scattering processes within materials to decouple the transport of phonons and electrons. In half-Heusler (hH) compounds, selective defect reduction can significantly improve performance due to the weak electron-acoustic phonon interaction. This study utilized Sb-pressure controlled annealing process to modulate the microstructure and point defects of $Nb_{0.55}Ta_{0.40}Ti_{0.05}FeSb$ compound, resulting in a 100% increase in carrier mobility and a maximum power factor of $78\,\mu W\,cm^{-1}\,K^{-2}$, approaching the theoretical prediction for NbFeSb single crystal. This approach yielded the highest average $zT$ of ~0.86 among hH in the temperature range of 300-873 K. The use of this material led to a 210% enhancement in cooling power density compared to $Bi_2Te_3$-based devices and a conversion efficiency of 12%. These results demonstrate a promising strategy for optimizing hH materials for near-room-temperature thermoelectric applications.

Thermoelectric (TE) is a solid-state energy conversion technology that enables the conversion between thermal and electrical energy[1–3]. Electrons near the Fermi level in TE material carry both charge and entropy and can migrate under the driving electric field or temperature field to realize the energy transfer. However, in this process, a portion of energy will inevitably be lost through heat conduction due to the temperature gradient ($\Delta T$) in the material. Therefore, the efficiency of a TE device is lower than a Carnot engine and depends primarily on the TE dimensionless figure of merit ($zT$), where $zT = [S^2\sigma/(\kappa_L + \kappa_e)]T$; $S$, $\sigma$, $\kappa_L$, $\kappa_e$, and $T$ are the Seebeck coefficient, electrical conductivity, lattice and electronic contribution of thermal conductivity, and absolute temperature, respectively. Thermoelectric performance is the result of a competition between electrical (power factor, $PF = S^2\sigma$) and thermal transport ($\kappa_L + \kappa_e$), both of which are heavily affected by scattering processes within the material[4–6]. In recent years, to improve TE performance, researchers have pursued reducing $\kappa_L$ by introducing various scattering sources, such as point defects[7,8], grain boundaries[9], dislocations[10,11], nanoprecipitations[12], and porous structures[13,14]. However, adding scattering centers to suppress phonon propagation also weakens the electrical transport, such as

reduced carrier mobility, which results in low PF and $zT$. In this scenario, balancing the beneficial and detrimental effects of the scattering source on TE transport becomes the essential strategy for optimizing TE properties. Thus, the ratio of carrier mobility to lattice thermal conductivity (i.e., $\mu/\kappa_L$) is critically important for enhancing TE performance[15]. On the other hand, mobility enhancement can be achieved by suppressing collisions of charged carriers with different scattering sources, including phonons, ionized impurities, defects, electromagnetic fields, and other carriers[4]. Among them, structural defects in TE materials play a major role at low temperatures, so minimizing contamination and reducing defect density is crucial for achieving high TE performance near room temperature[16].

Half-Heusler (hH) alloys are among the most promising TE materials for medium and high-temperature waste heat recovery applications owing to their outstanding mechanical strength, thermal stability, and $zT$[17,18]. Due to the suppressed electron-acoustic phonon coupling in hH alloys, electron scattering from phonons is much weaker with the intrinsic low deformation potential[19]. Therefore, defects in hH alloys play a more important role in electronic transport than that in other TE materials[20]. It is possible to

[1]Department of Materials Science and Engineering, Pennsylvania State University, University Park, PA 16802, USA. [2]Present address: Beijing National Laboratory for Condensed Matter Physics, Institute of Physics, Chinese Academy of Sciences, Beijing 100190, China. [3]These authors contributed equally: Hangtian Zhu, Wenjie Li. ✉e-mail: htzhu@iphy.ac.cn; wzl175@psu.edu; sup103@psu.edu; bup346@psu.edu

enhance mobility by reducing the scattering source, such as lattice defects in the hH materials, thereby achieving high PF. In this study, we demonstrate significantly enhanced TE performance of $M_{1-x}Ti_xFeSb$ (M = Nb, Ta) compounds, one of the most promising p-type hH material families[21–24], by reducing point defect density. The Nb\Ta ratio was tuned to improve the $\mu/\kappa_L$ ratio, as it can introduce mass fluctuation without lattice stress (Fig. S1). This is due to the small difference in lattice constant of less than 1% between NbFeSb and TaFeSb[24]. We developed an Sb-pressure controlled annealing process to reduce the concentration of defects and drive grain growth (Fig. 1a). The microstructure shows sub-millimeter-sized grains in p-type $Nb_{0.55}Ta_{0.40}Ti_{0.05}FeSb$ sample with reduced grain boundary density (Fig. 1b). The minimized defects and grain boundary density enable substantial enhancement of mobility (Fig. 1a) and thus double the PF up to ~78 μW cm⁻¹ K⁻² near room temperature (Fig. 1c), which leads to the highest $zT$ of ~0.4 at room temperature in hH alloys and the highest average $zT$ of 0.86 in the temperature range of 300–873 K among hH. This enhancement is reflected in an excellent power conversion efficiency of ~12% under $\Delta T$ of 600 K.

In the field of TE cooling technology, two important aspects are the maximum temperature difference ($\Delta T$) and the maximum cooling density under zero $\Delta T$, which depend on specific application requirements. Over the past six decades, the maximum $\Delta T$ has been established on $Bi_2Te_3$-based alloys due to their high $zT$ near room temperature, which has yet to be significantly rivaled. However, for most TE applications that require small $\Delta T$ and precise temperature control, such as in electronics temperature control, and particularly for hot-spots cooling[25,26], the maximum cooling density under small $\Delta T$ is often more important. For instance, in laser diode cooling, where a significant amount of heat is generated during operation and must be maintained at a specific temperature for optimal performance and

longevity, the maximum cooling heat flow density ($Q_{c_{max}}$) is of great importance, and it is governed by the relation[27]:

$$Q_{c_{max}} \sim \frac{1}{2} PF \cdot T_c^2 - \Delta T \cdot \kappa \qquad (1)$$

A higher $zT$ value does not necessarily result in higher heat pumping, as it is not directly related to $Q_{c_{max}}$. Therefore, in order to achieve better $Q_{c_{max}}$ performance, it is important to have a large PF, especially when the $\Delta T$ is small. Under extreme conditions when $\Delta T$ is negative, a high PF and $\kappa$ can further enhance heat pumping by combining active and passive cooling[28]. Currently, most TE cooling devices use $Bi_2Te_3$ alloys and provide a small $\Delta T$ of 5–10 K for high-power electronics thermal management. For the first time, we demonstrate superior solid-state cooling power density in hH alloys that outperforms the state-of-the-art $Bi_2Te_3$-based modules. We have fabricated a uni-couple device using $YbAl_3$ as an n-type counterpart, which exhibits ~210% higher cooling density with a $\Delta T$ of 5 K compared to a $Bi_2Te_3$-based device (Fig. 1d). This performance is highly relevant for high-power electronics thermal management such as laser sources and on-chip hot-spot.

## Results and discussion
### High-performance hH materials enabled by point defect and grain boundary density modulation

Due to the weak acoustic scattering in hH alloy and the screening effect of free electrons, an increase in carrier concentration drives the competition between optical and acoustic phonon scattering, which results in a peak in carrier mobility[29]. For the p-type (Nb,Ta)FeSb matrix, which has a heavier density of state effective mass of valence band ($m_d^* \sim 8m_e$)[6,22,23], an optimized light doping strategy of 5% Ti at Nb/Ta site has been developed (Fig. S2). While ionized impurities can

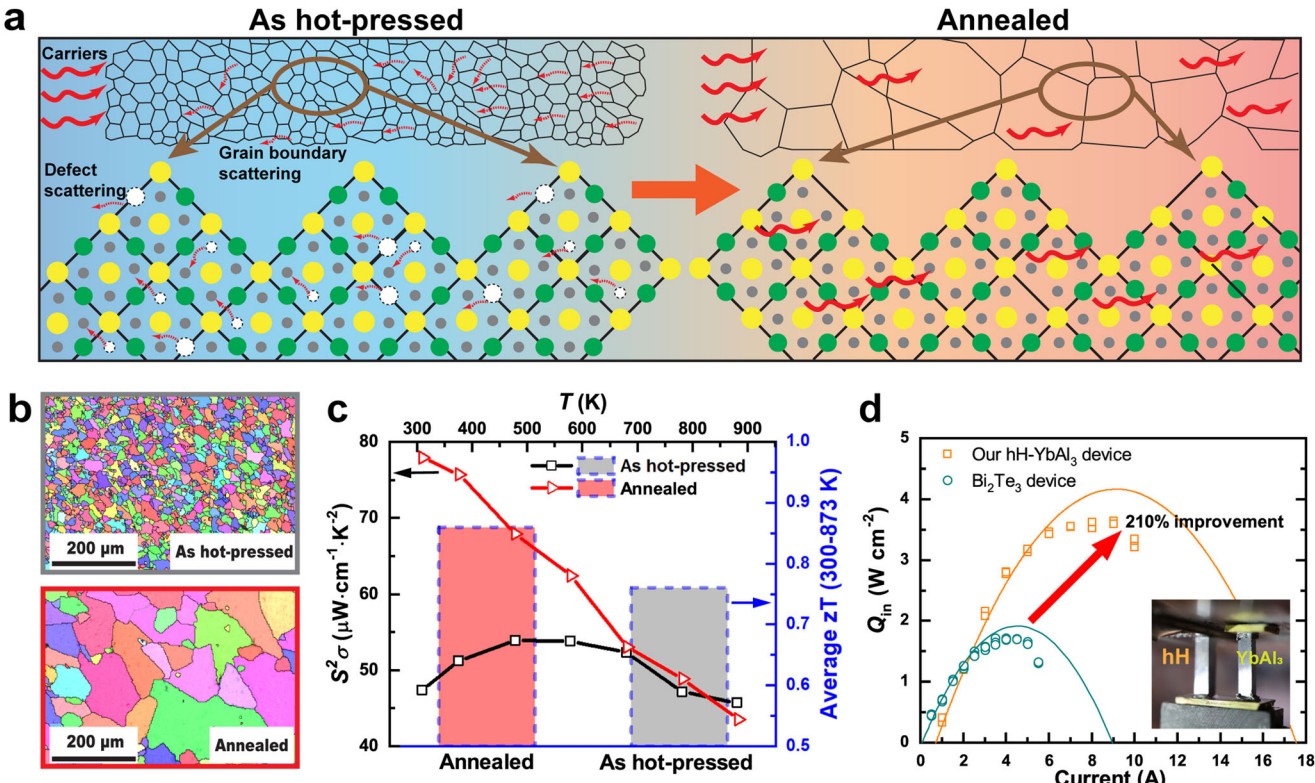

**Fig. 1 | High performance of $Nb_{0.55}Ta_{0.4}Ti_{0.05}FeSb$ hH material and devices with sub-millimeter grains. a** Schematic illustration of enhanced carrier transportation by Sb-pressure controlled annealing. Green: Nb/Ta/Ti; Gray: Fe; Yellow: Sb; White: point defects. **b** Electron backscattered diffractometry (EBSD) images of as-hot-pressed and annealed alloy. **c** Power factor as a function of temperature and $zT$ near room temperature. **d** Cooling power of hH-YbAl₃ uni-couple device and $Bi_2Te_3$-based device at $\Delta T$ of 5 K.

significantly influence carrier mobility, the improvement in mobility by reducing doping concentration is still limited. Microstructural features, such as grain architecture and crystal quality, are strongly correlated with defect density and depend on the fabrication process. These features govern the electrical and thermal transports by acting as scattering sources. Fast powder consolidation typically leads to uniform polycrystalline samples with grain sizes of hundreds of nanometers[9,23,30], while annealing above the recrystallization temperature can lower dislocation density and the number of grain boundaries due to the grain growth process. However, the large grain size in hH samples is difficult to achieve by annealing[22,31], owning to the alloys' very high melting temperature (~1273 K) and good thermal stability, even though grain growth in hot-pressed nanocrystals is thermodynamically favorable due to the reduction in surface energy.

By adding addition Ta into the $Nb_{0.55}Ta_{0.40}Ti_{0.05}FeSb$ compound, its phase transition temperature decreases to 1173 K. Consequently, the hot-pressing for the $Nb_{0.55}Ta_{0.40}Ti_{0.05}FeSb$ sample can only be conducted at 1073 K, which is approximately 300 K lower than that of the NbFeSb compound, and no remarkable grain growth was observed during the hot-pressing process (Fig. 2a). Even after conventional vacuum annealing for 6 days at 1123 K, the grain size of the as-hot-pressed nano-crystalline $Nb_{0.55}Ta_{0.40}Ti_{0.05}FeSb$ sample did not increase significantly (Fig. S3). TEM investigation revealed (Fig. S4a–h) that the pinning behavior of the Sb-rich phase imparts dynamic resistance to grain growth at the grain boundary. To remove the excess Sb and provide an Sb-deficient environment, an Sb-pressure-controlled annealing process was developed under different temperatures (see Supplementary Information for details). As a result, no Sb-rich phase was observed at grain boundaries after Sb-pressure controlled annealing at 1143 K for 2 days (Fig. S4k, l), and grain sizes were significantly increased by three orders of magnitude, from ~200 nm to ~150 μm (Fig. 2a–d), due to the substantially improved grain boundary

migration rate (see Supplementary Information for details). The vacancy in the lattice of the samples, where internal energy is stored during the ball-milling and hot-pressing process, was also minimized after both types of annealing processes, as confirmed by SEM and TEM results (Fig. 2e–h and Fig. S4i, j).

Figure 3a–c provides insight into the effect of Sb-pressure controlled annealing on electron transport in the $Nb_{0.55}Ta_{0.40}Ti_{0.05}FeSb$ compound. The electrical conductivity is substantially improved due to the enhanced carrier mobility resulting from the reduced point defects and grain boundary density as the grain size is increased during the Sb-pressure controlled annealing process. The observed $T^{-1.5}$ trend of the electrical conductivity suggests that the scattering of defects is minimized, indicating the dominance of acoustic phonon scattering in the transport behavior of the sample annealed at high temperature with controlled Sb-pressure (Fig. 3a). To distinguish the contribution of scattering from point defect and grain boundary, we compared the samples before and after conventional vacuum annealing, where the point defects are released with the unchanged grain size. The increase in mobility in the conventional vacuum-annealed sample can thus be attributed to the reduction of a point defect. In comparison, the elimination of point defects contributes ~25% of the total increase in carrier mobility (the first stage in Fig. 3c), while the reduction of grain boundaries contributes up to ~75% of the total increase (the second stage in Fig. 3c), indicating the strong scattering effect of the potential barrier at grain boundaries in hH system[20]. Thus, a large PF of ~78 μW cm$^{-1}$ K$^{-2}$ near room temperature is achieved (Fig. 3b), which is two times higher than that of the as-hot-pressed sample and close to the theoretical optimum PF (~90 μW cm$^{-1}$ K$^{-2}$) for NbFeSb single crystal[19].

The mean free path ($\lambda$) of phonon in hH is typically less than 100 nm[19], which is less than the grain size of materials investigated in this study. Consequently, increasing the grain size did not significantly

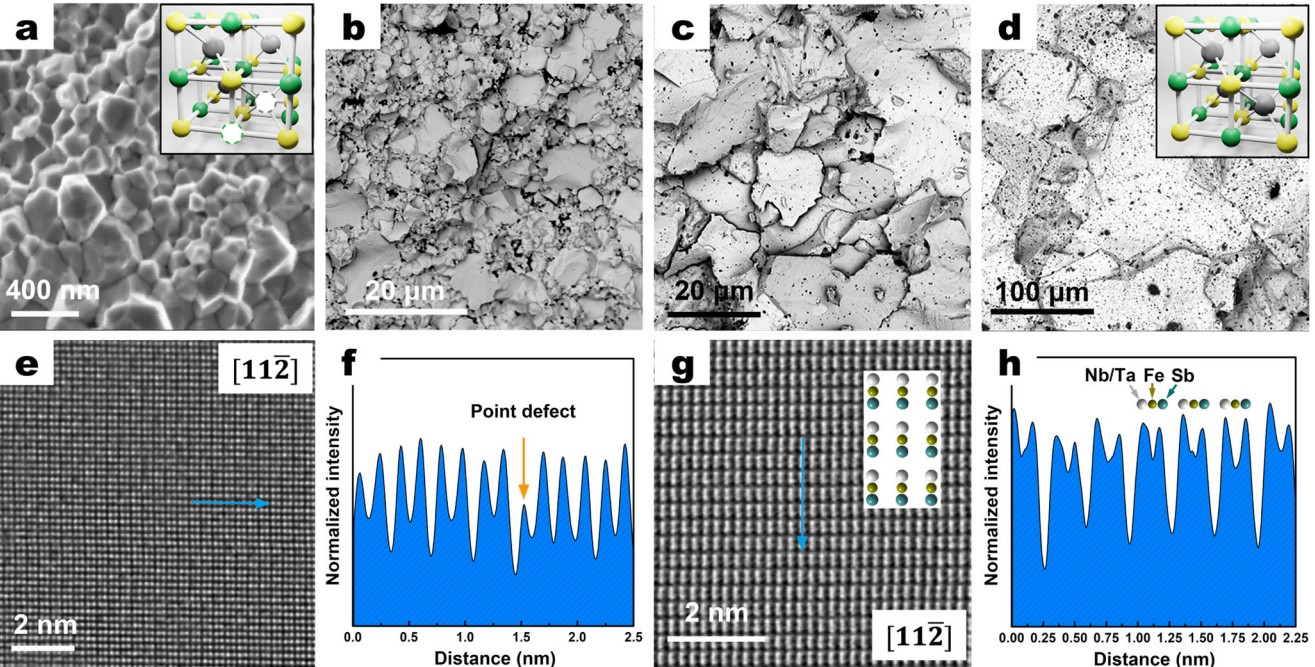

**Fig. 2 | Growth of grain in Nb0.55Ta0.40Ti0.05FeSb samples with Sb-pressure control. a–d** The SEM images of samples under different heat treatment conditions: as-hot-pressed at 1123 K for 5 min consisting of grain size of 200–300 nm (average ~250 nm) (**a**), annealing with Sb-pressure control for 2 days at 1073 K consisting of grain size of 2–10 μm (average ~6 μm) (**b**), at 1113 K exhibiting grain size of 5–30 μm (average ~20 μm) (**c**), and at 1143 K exhibiting grain size of 20–200 μm (average ~150 μm) (**d**). The crystal structures are inserted in (**a**) and (**d**),

where green, gray, yellow, and white indicate the Nb/Ta/Ti, Fe, Sb, and point defects of vacancies. **e–h** HRTEM images along [11$\bar{2}$] axis and defect analysis of the as-hot-pressed sample (**e, f**) and the sample annealed at 1143 K with Sb-pressure control (**g, h**). The actual Wyckoff positions of vacancy are unable to be distinguished due to the similar atomic weight of the two sites ($Nb_{0.55}Ta_{0.4}Ti_{0.05}$ = 125.8710, Sb = 121.7600).

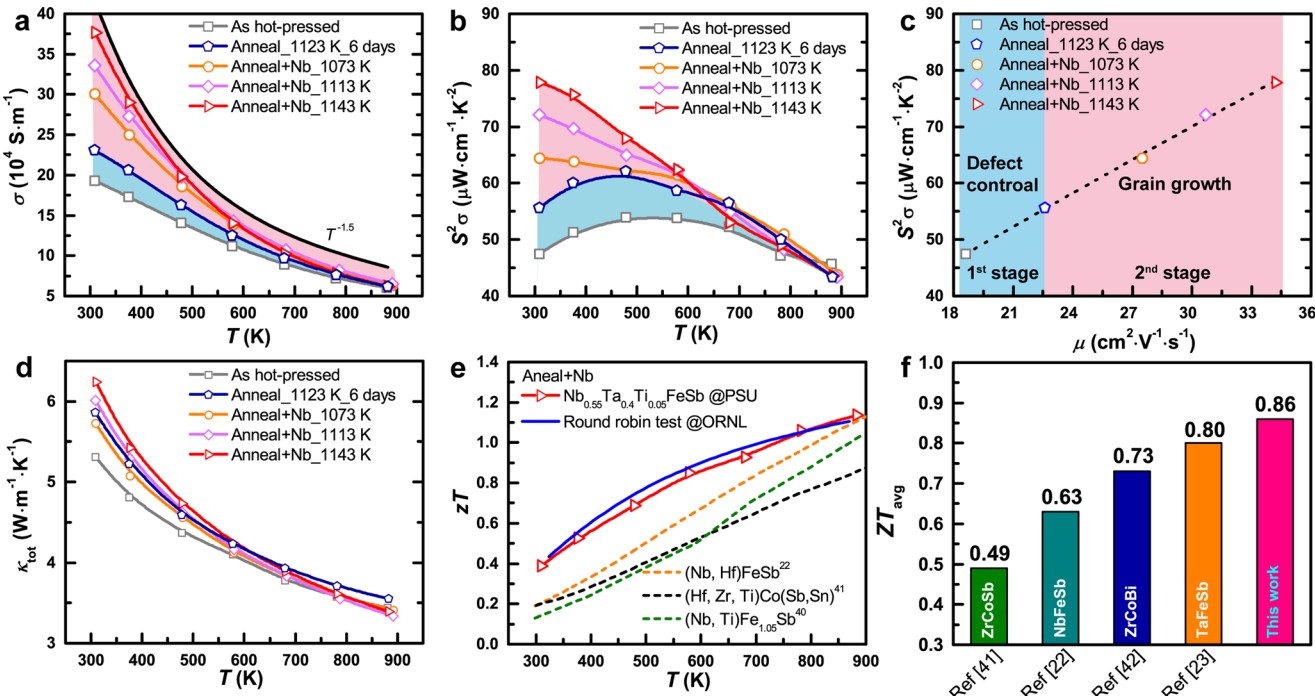

**Fig. 3 | Enhanced TE performance of (Nb,Ta,Ti)FeSb sample. a, b** The electrical conductivity (**a**) and PF (**b**) as a function of temperature under different annealing conditions. **c** The carrier mobility-dependent PF under different annealing conditions near room temperature. The contribution of optimization of point defects and crystal boundary for the mobility enhancement is marked by blue and pink in (**a–c**), respectively. **d, e** The temperature-dependent thermal conductivity (**d**) and $zT$ values (**e**). **f** The average $zT$ compared with state-of-the-art p-type hH material[22,23,40–42].

improve the lattice thermal conductivity $\kappa_L$ due to the weak phonon scattering from the grain boundary[15], which has also been observed in $Mg_3(Bi,Sb)_2$ system[32,33]. As a result, this led to a substantially increased $\mu/\kappa_L$ ratio, resulting in ~200% enhanced $zT$ value near room temperature compared to other hH materials (Fig. 3e and Fig. S5). Remarkably, a record-high average $zT$ of ~0.86 was achieved from room temperature to 873 K (Fig. 3e, f). The reproducibility of the performance $Nb_{0.55}Ta_{0.40}Ti_{0.05}FeSb$ sample was verified in Fig. S6, and the round-robin test was also conducted at Oak Ridge national laboratory (Fig. S7), which confirmed the transport data reported here.

## High cooling capacity hH module performance

The $Bi_2Te_3$-based materials have been widely used in commercial solid-state cooling devices with a cooling power density of up to ~45 W cm⁻² (based on leg lengths of ~0.2 mm, Table S1)[34]. However, it is not sufficient for advanced cooling applications, such as electronic cooling and thermal management for laser devices and hot-spot, which require higher cooling power densities up to ~100 W cm⁻². The most crucial parameter for high-power TE cooling is PF, which determines the maximum cooling power density of the module ($Q_{c_{max}}$) as:

$$Q_{c_{max}} = \frac{(T_c^2 - 2\Delta T/z)\text{PF}}{2l} = \frac{(\text{PF}T_c^2 - 2\Delta T\kappa)}{2l} \quad (2)$$

where $T_c$ is the cold side temperature, and $l$ is the length of the leg. For the materials with the same $z$ ($\text{PF}/\kappa_{tol}$) value, $Q_{c_{max}}$ is mainly determined by PF, particularly under small $\Delta T$. For instance, as shown in Fig. 4a, under a $\Delta T$ of 5 K, $Q_{c_{max}}$ shows a linear relationship with PF, while the effect of $\kappa$ is almost negligible. When PF is doubled (e.g., from ~35 μW cm⁻¹K⁻² for $Bi_2Te_3$-based materials to ~80 μW cm⁻¹K⁻² for the current hH materials), $Q_{c_{max}}$ is enhanced by ~250%. The effect of $z$ value on $Q_{c_{max}}$ and the effect of performance (COP) corresponding to $Q_{c_{max}}$ gradually decreases with a decrease of $\Delta T$ (Fig. S8). It is also worth noting that the maximum cooling capacity of a device, $Q_{c_{max}} = \frac{\text{PF}T_c^2}{2l}$, is

directly proportional to the PF of the material where $\Delta T$ is equal to 0. The parameters of p-type $Nb_{0.55}Ta_{0.40}Ti_{0.05}FeSb$ and p-type $Bi_{0.33}Sb_{1.67}Te_3$ materials considered in Eq. 2 are listed in Table S2 for reference.

Benefiting from the ultra-high PF and the decent $z$ value near room temperature, $Nb_{0.55}Ta_{0.40}Ti_{0.05}FeSb$ exhibits enhanced performance compared to that of $Bi_2Te_3$-based TE materials in terms of low $\Delta T$ heat pumping. The $YbAl_3$ alloy with an extremely high PF of ~110 μW cm⁻¹K⁻² at room temperature (Fig. S9), which is compatible with p-type $Nb_{0.55}Ta_{0.40}Ti_{0.05}FeSb$, was used as the n-type counterpart to construct the uni-couple cooling device (hH-YbAl₃). Figure 4b shows the simulated cooling performance of hH-YbAl₃ device. Considering the extremely high electrical and thermal conductivity of $YbAl_3$, the cross-section of the $YbAl_3$ leg should be four times smaller than that of p-type hH to optimize the performance of the device, which makes the construction and measurement of the device extremely difficult. Thus, the cross-section ratio ($A_p/A_n$) was reduced to 1.4, leading to slightly lower performance than that of the device with optimized geometry. The measured $Q_{c_{max}}$ of the hH-YbAl₃ device at $\Delta T$ of 5 K was found to be 3.6 W cm⁻² (with a leg length of 7.7 mm), which is twice that of the $Bi_2Te_3$-based device, 1.7 W cm⁻², with the same leg length and $\Delta T$ (Fig. 4b, c and Fig. S10). The maximum cooling power density of the hH-YbAl₃ device remains competitive with that of $Bi_2Te_3$-based devices up to $\Delta T$ of 20 K. Improved n-type material would be able to provide even higher performance at high $\Delta T$. For example, a hypothetical device made using both legs with similar properties to $Nb_{0.55}Ta_{0.40}Ti_{0.05}FeSb$ (which has a negative Seebeck coefficient for the n-type leg) would result in a higher cooling capacity across a wider $\Delta T$ range of 46 K (Fig. 4b).

The COP of the hH-YbAl₃ device, which is determined by the $zT$ value, $\Delta T$, and current $I$, is demonstrated in Fig. 4d. The current applied to the device determines heat load, as shown in Fig. 4c. Practically when TE cooling devices work under a high heat load, the COP is usually lower than its maximum value at low cooling density (Fig. S10c,

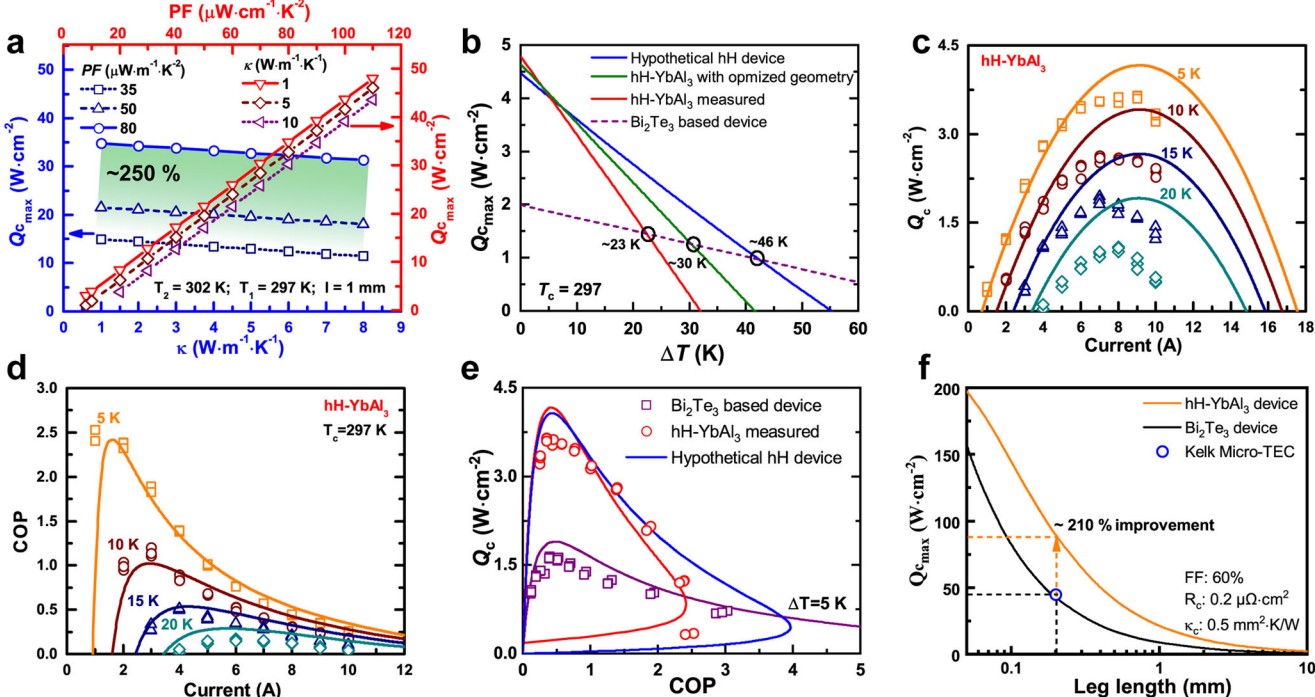

**Fig. 4 | Theoretical and experimental cooling performance of the hH-YbAl3 TE device. a** The theoretical maximum $Q_c$ as a function of PF (under different $\kappa$) and $\kappa$ (under different PF) calculated using Eq. 2. **b** The cooling $\Delta T$ dependent $Q_{c_{max}}$ of different hH-YbAl$_3$ devices. **c, d** The calculated and measured cooling power density (**c**) and COP (**d**) of hH-YbAl$_3$ device with different $\Delta T$. **e** The calculated and measured cooling density as a function of COP at $\Delta T$ of 5 K. The leg height of hH-YbAl$_3$ and Bi$_2$Te$_3$-based modules are the same at 7.7 mm. The details of Bi$_2$Te$_3$-based materials and device is described in Figs. S10 and S11, and Table S2. The hypothetical hH device is calculated using the same TE properties of Nb$_{0.55}$Ta$_{0.40}$Ti$_{0.05}$FeSb for both n- and p-type legs. **f** The calculated projection of $Q_{c_{max}}$ of hH-YbAl$_3$ module under different leg heights as a comparison with a state-of-the-art commercial cooling device. The fill fraction (FF), electrical ($R_c$), and thermal ($\kappa_c$) contact resistance used for calculation are listed in the figure.

d)[35]. Under small $\Delta T$, the hH-YbAl$_3$ device demonstrates a significantly higher cooling density ($Q_c$) compared to that of a traditional Bi$_2$Te$_3$-based module under the same COP up to the magnitude of 2 (Fig. 4e). It also exhibits a slower COP degradation trend with an increase of cooling density as compared to that of Bi$_2$Te$_3$-based device[36]. The maximum $Q_{c_{max}}$ of ~88 W cm$^{-2}$ can be achieved for the hH-YbAl$_3$ device when the length of the TE leg is projected to be 0.2 mm (Fig. 4f), which is ~210% higher than that of commercial Bi$_2$Te$_3$-based modules. This illustrates that hH-YbAl$_3$ devices are more effective under small $\Delta T$ (e.g., 5 K).

## High efficiency for power generation

The outstanding performance of the developed hH material with a high average $zT$ has also been verified by the device power generation mode. A high conversion efficiency of ~12%, which is ~40% higher compared to our previous work[37], was achieved at a hot-side temperature of 923 K for the hH uni-couple module (Fig. 5). This result is due to the superior performance of p-type hH leg with high average $zT$. The agreement between prediction and experimental internal resistance ($R_i$) and open voltage ($V_{oc}$) of uni-couple devices indicates that the experimental electrical contact resistance is negligible since the electrical contact resistance was not considered in the prediction (Fig. 5b, c). As shown in Fig. 5d, the slightly higher experimental $Q_{cond}$ compared to that of prediction results from the thermal radiation effect, which was observed in our previous work[38]. Thus, thermal radiation calibration was performed using a dummy module for accurate efficiency measurement (see Supplementary Materials and Fig. S12). During the three cycles of measurement, the performance of the device was stable and consistent with theoretical prediction, indicating good stability of the device and accuracy of the measurement (Fig. 5e, f). It is worth noting that the output power of the hH uni-couple module is also higher than

that of the Nb$_{0.95}$Ti$_{0.05}$FeSb/Zr$_{0.44}$Hf$_{0.44}$Ti$_{0.12}$NiSn$_{0.9}$Sb$_{0.01}$ module developed in our previous work, even though the PF of Nb$_{0.95}$Ti$_{0.05}$FeSb compound near room temperature (~100 µW cm$^{-1}$ K$^{-2}$) is higher than that of our p-type Nb$_{0.55}$Ta$_{0.40}$Ti$_{0.05}$FeSb compound. This result is mainly attributed to the better matching of electrical conductivity and thermal conductivity between the p-Nb$_{0.55}$Ta$_{0.40}$Ti$_{0.05}$FeSb and n-Zr$_{0.44}$Hf$_{0.44}$Ti$_{0.12}$NiSn$_{0.9}$Sb$_{0.01}$ (Fig. S13).

In summary, by minimizing chemical vacancies and reducing grain boundaries, the superior enhancement in power factor and average $zT$ value in Nb$_{0.55}$Ta$_{0.40}$Ti$_{0.05}$FeSb has been achieved. It was accomplished by controlling the Sb pressure during annealing to promote the growth of large grain size and reduce defects. The power factor of the materials was greatly improved by doubling room temperature carrier mobility and optimizing dopant concentration leading to the remarkable near-room temperature $zT$ of ~0.4 and the highest average $zT$ of ~0.86 (300–873 K) for hH TE materials. A 210% higher cooling capacity was achieved compared to that of the Bi$_2$Te$_3$ device under $\Delta T$ of 5 K by deploying the high PF hH material. The hH-YbAl$_3$ device has the advantage of maximum cooling density with $\Delta T \leq 20$ K and is even more efficient under low $\Delta T$ of 5 K and high cooling power density of 1-3.5 W cm$^{-2}$. In the power generation mode, the measured energy conversion efficiency of the uni-couple hH module reached ~12% at $\Delta T$ of 600 K. This work successfully demonstrates the effectiveness of low-defect design in improving hH performance and the great potential of Nb$_{0.55}$Ta$_{0.40}$Ti$_{0.05}$FeSb in high heat flux thermal management.

## Methods
### Synthesis of hH materials
The p-type (Nb,Ta,V,Ti)FeSb alloys were synthesized by ball milling. The niobium (99.9%, foil), tantalum (99.9%, shot), vanadium (99.99%, piece), titanium (99.9%, sponge), iron (99.99%, slug), and antimony

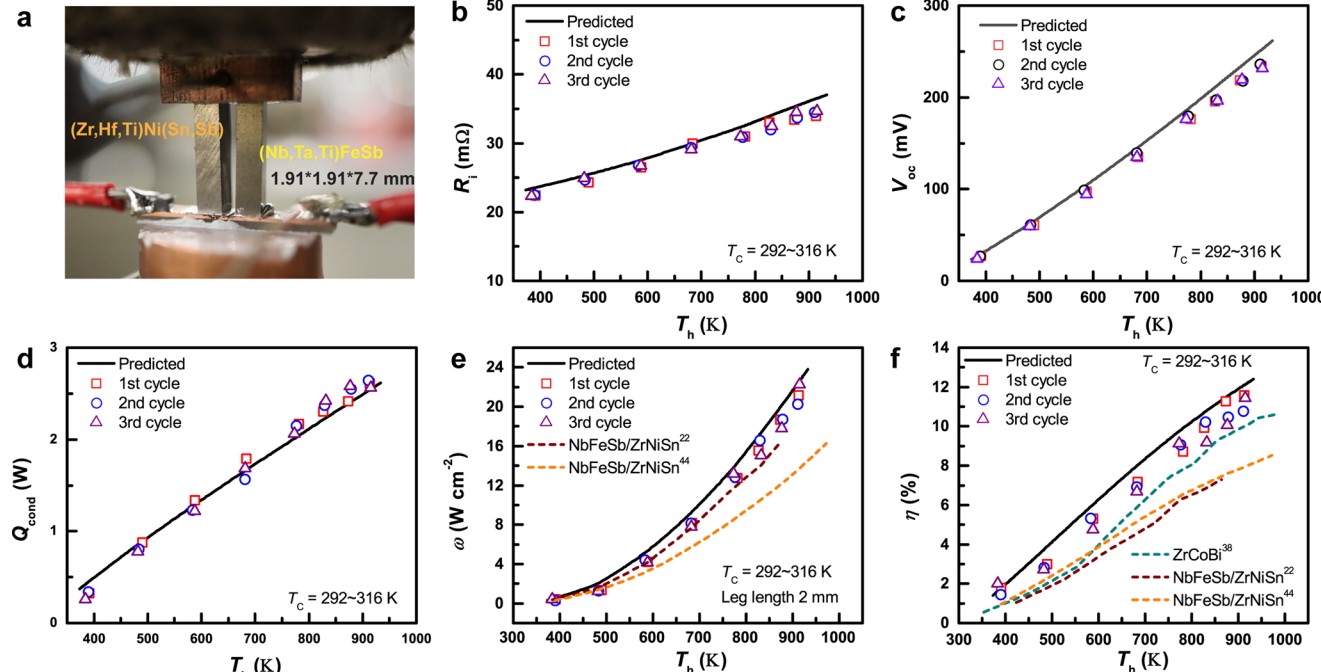

**Fig. 5 | Power generation performance of hH devices. a** Photo of the measurement setup for the high-performance hH uni-couple module. **b–f** Internal resistance ($R_i$) (**b**), open voltage ($V_{oc}$) (**c**), heat flow at zero current ($Q_{cond}$) (**d**), the power density of TE legs ($\omega$) (normalized to the length of 2 mm) (**e**), and conversion efficiency ($\eta$) (**f**) obtained using optimal current as a function of hot side temperature ($T_h$)[22,42–44]. The radiation in the heat flow measurement was calibrated by the standard sample[38] (Fig. S12).

(99.9999%, shot) constituents were weighted in their stoichiometric ratio inside the glove box. Elements with a total mass of 15 g were loaded into a stainless-steel ball-milling jar in a glove box under an Ar atmosphere. The high-energy ball milling was conducted for 24 hours using SPEX mixer/mill (Model 8000D, SPEX SamplePrep, Metuchen, NJ). The ball-milled powders were consolidated by SPS (SPS, Dr. Sinter-625V, Fuji, Japan) at 1073 K under a pressure of 80 MPa for 5 min. The hot-pressed pellets were sealed with Nb foil in quartz tubes under a high vacuum. The Nb foil was used to control the Sb pressure during the annealing.

The n-type $Zr_{0.44}Hf_{0.44}Ti_{0.12}NiSn_{0.9}Sb_{0.01}$ materials were synthesized by induction melting under an argon atmosphere[38]. The zirconium (99.9%, slug), hafnium (99.9%, piece), titanium (99.9%, wire), nickel (99.995%, slug), tin (99.95%, wire), and antimony (99.9999%, shot) constituents were weighted in their stoichiometric ratio and loaded into tungsten crucible. The induction melting was performed under an argon atmosphere for 3 min and re-melted 3 times to ensure homogeneity. Then the ingots were transferred into stainless steel jar inside the glove box. The high-energy ball milling was conducted for 4 h using SPEX mixer/mill (Model 8000D, SPEX SamplePrep, Metuchen, NJ). The grinded nanopowders were consolidated by SPS at 1373 K under a pressure of 80 MPa for 5 min.

### Synthesis of YbAl₃ materials

The YbAl₃ materials were synthesized by induction melting under an argon atmosphere. The mixtures of the high-purity metals of ytterbium (99.9%, slug) and aluminum (99.99%, slug) with a nominal composition of YbAl₃.₂ were loaded into a tungsten crucible. The induction melting was performed under an argon atmosphere for 3 min and re-melted 3 times to ensure homogeneity. Then the ingots were ball-milled for 4 h, and the grinded nanopowders were consolidated by SPS at 773 K under a pressure of 80 MPa for 5 min. The obtained pellets were sealed in quartz tubes under high vacuum and annealed at 973 K for 30 min.

### Synthesis of Bi₂Te₃-based materials

The p-type $Bi_{0.33}Sb_{1.67}Te_3$ and n-type $Bi_2Te_{2.7}Se_{0.3}$ were synthesized by the high energy ball milling method. The antimony (99.9%, shot), bismuth (99.9%, pieces), tellurium (99.9%, lumps), and selenium (99.9%, pellets) constituents were weighted in their stoichiometric ratios inside the glovebox, and sealed in a sealed quartz tube and melted for 24 hours at 1000 °C inside a conventional furnace following by quenching. The ingot was then loaded into a hardened steel vial along with steel balls with the ball to powder ratio of 1:1 and ball milled for 3 hours using a SPEX mixer/mill (Model 8000D, SPEX SamplePrep) to obtain a homogeneous powder. The powders were then consolidated in a cylindrical graphite die using spark plasma sintering (SPS) at 520 °C (for p-type) and 450 °C (for n-type) under constant 30 MPa pressure for 2 min.

### Measurement of thermoelectrical (TE) properties

The electrical conductivity and Seebeck coefficient were measured simultaneously (ULVAC-RIKO ZEM-3 system, Japan). High-temperature thermal properties were determined by measuring thermal diffusivity with a laser flash system (LFA-467 HT Hyper-Flash®, Germany). Specific heat was measured with a differential scanning calorimeter (Netzsch DSC 214, Germany). The thermal conductivity, $\kappa$, was calculated from $\kappa = \alpha \cdot C_p \cdot \rho$, where $\alpha$, $\rho$, and $C_p$ are thermal diffusivity, density, and specific heat. The density is measured using the Archimedes method. The uncertainties in electrical conductivity, thermal conductivity, Seebeck coefficient, and $zT$ are ±5, ±2, ±5, and ±7%, respectively. The microstructure of alloys and uni couple module junctions is characterized by field emission scanning electron microscopy (FESEM, FEI Verios G4), energy dispersive spectroscopy (EDS, Oxford Aztec), and electron backscattered diffractometry (EBSD, FEI Apero S, Oxford AztecCrystal). The carrier density and mobility were measured by LakeShore Hall Effect System (8400 Series HMS, LakeShore). The TEM is performed using FEI Titan G2.

## The module preparation

The p-type $Nb_{0.55}Ta_{0.40}Ti_{0.05}FeSb$ leg with a height of 7.7 mm and cross-section of 3.6 mm$^2$ was used to fabricate both power generation and cooling devices. The $Zr_{0.44}Hf_{0.44}Ti_{0.12}NiSn_{0.9}Sb_{0.01}$ material with the same height (7.7 mm) and cross-section (3.6 mm$^2$) was used for the n-type leg of power generation uni-couple device. The n-type $YbAl_3$ leg with a height of 7.7 mm and a cross-section of 2.53 mm$^2$ was used for the cooling device. With the aim of minimizing the relative error from heat flow measurement, the cross-section $Bi_2Te_3$ legs were adjusted to match the maximum heat flow of the hH-$YbAl_3$ device at $\Delta T$ of 5 K. Therefore, p-type $Bi_{0.33}Sb_{1.67}Te_3$ leg (height of 7.7 mm and cross-section of 6.4 mm$^2$) and n-type $Bi_2Te_{2.7}Se_{0.3}$ leg (height of 7.7 mm and cross-section of 7 mm$^2$) were used to fabricate the $Bi_2Te_3$ based cooling device.

AlN-based direct bonding copper (DBC) and copper plate were used as the substrate of the module. Ga-based liquid metal, which could provide extremely low electrical and thermal contact resistance, was used as the interfacial material between the hH legs and the substrate for the TE module. Due to the reaction between $YbAl_3$ and liquid metal, $YbAl_3$ legs were connected to the substrate by reflow soldering with Al–Zn-based solder at 673 K (Fig. S14). The custom-built scanning probe measurement system was used to analyze the contact resistance[37–39].

## Heat flow measurement

The heat flow ($Q$) was measured using a Q-meter, which is a standard material (copper, brass, or graphite are commonly used) with a cylindrical or rectangular shape connected to the heat sink. The temperature gradient along the Q-meter (copper and graphite were used for power generation and cooling measurement in this work, respectively) was measured by using four thermocouples located at specific distances[39]. Then $Q$ can be calculated by $Q = \kappa \cdot A \cdot \frac{dT}{dx}$, where $\kappa$, $A$, and $dT/dx$ are the thermal conductivity, cross-sectional area, and the slope of temperature difference versus distance on the Q-meter.

## The thermal radiation calibration

In the process of power generation measurement with hot-side temperature up to 923 K, the Q-meter absorbs the radiation energy from the high-temperature heater, resulting in the heat flow measurement error. In order to calibrate the thermal radiation, a dummy uni-couple device made of two legs of the $Zr_{0.44}Hf_{0.44}Ti_{0.12}NiSn_{0.9}Sb_{0.01}$ material was fabricated and measured. The thermocouples were soldered on the DBC substrates to measure the temperature on both sides of the leg. The thermal flow was calculated based on the thermal conductivity of $Zr_{0.44}Hf_{0.44}Ti_{0.12}NiSn_{0.9}Sb_{0.01}$ material and the temperature of two sides of the leg (Fig. S12).

## Data availability

All data are available in the main text and the Supplementary Materials.

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

## Acknowledgements

**Funding.** We acknowledge Dr. Hsing Wang for the round-robin test of thermoelectric materials and Kai Wang for the crystal structure drawing. H.Z. acknowledges the financial support through the Office of Defense Advanced Research Projects Agency (DARPA) under the project of Nano Engineered Thermoelectric Systems (NETS). W.L. and B.P. acknowledge the financial support through the Office of Naval Research through grant number N00014-20-1-2602. This effort (A.N. and S.P.), including the data analysis and interpretation, was supported as part of the Center for 3D Ferroelectric Microelectronics (3DFeM), an Energy Frontier Research Center funded by the U.S. Department of Energy (DOE), Office of Science, Basic Energy Sciences under Award Number DE-SC0021118. N.L. acknowledges the support from National Science Foundation (NSF) planning grant, TERRM. Y.Z. acknowledges the support from the Army SBIR program supported by NanoOhmics.

**Legal disclaimer.** This report was prepared as an account of work sponsored by an agency of the United States Government. Neither the United States Government nor any agency thereof, nor any of their employees, makes any warranty, express or implied, or assumes any legal liability or responsibility for the accuracy, completeness, or usefulness of any information, apparatus, product, or process disclosed, or represents that its use would not infringe privately owned rights. Reference herein to any specific commercial product, process, or service by trade name, trademark, manufacturer, or otherwise does not necessarily constitute or imply its endorsement, recommendation, or favoring by the United States Government dns of authors expressed herein do not necessarily state or reflect those of the United States Government or any agency thereof.

## Author contributions

H.Z. designed this work. H.Z. and W.L. synthesized the samples, performed the microstructure characterizations, measured the thermoelectric properties, assembled the modules, and measured module performance. A.N. and Y.Z. helped with the measurements of device performance and theoretical stimulation. N.L. helped with the microstructure characterizations and analysis. H.Z. and W.L. wrote the paper. All authors contributed to the data analysis and edited the paper. B.P. and S.P. supervised the research. All authors commented on, discussed, and edited the paper.

## Competing interests

The authors declare no competing interests.
