## [Peer Review File · Nature Communications]

Half-Heusler Alloys as Emerging High Power Density Thermoelectric Cooling MaterialsREVIEWER COMMENTS

Reviewer #1 (Remarks to the Author):

In this work, the authors fabricated HH materials and devices and then investigated the cooling and power generation efficiencies. They want to demonstrate that HH alloys are more suitable for cooling applications than bismuth telluride alloys by emphasizing that HH alloys have higher power factors, leading to higher cooling power density. Although the present work may be technically important, I am quite hesitating to recommend it to publication in Nature Communications because of the following concerns.

(1) The hH alloys have been well studied and the compositions focused in this work show very common ZT values.

(2) Most importantly, for cooling applications, the maximum temperature difference is another important parameter. For commercialized bismuth telluride modules, it can reach 70 degree C, but how about the present hH module?

(3) The authors made comparison between hH and Bi₂Te₃, but no detailed information can be found. The authors cited a reference from the same author group, but also no materials compositions and properties are reported in this paper.

(4) For the power generation, the efficiency is not high for such a large temperature difference. What are the electrode and contact layer used for the power generation devices?

(5) The authors spent a lot of spaces to emphasize the importance of grain growth, but it is also well known and the method has been reported.

(6) About Figure 2 e to h, it is written as "The vacancies in crystalline lattice, where the internal energy is accumulated during ball milling and hot-pressing process, were minimized after this process". Very unclear, more explanations are needed.

(7) In the abstract, the second sentence seems uncompleted.

(8) In the conclusion, the cooling power density is written as "1-3.5 W cm⁻²", much lower than that in the text, why?

Reviewer #2 (Remarks to the Author):

The publication on high power cooling performance of (Nb,Ta)FeSb is an interesting contribution to half-Heusler thermoelectric. The grain size of the half-Heusler materials is increased by orders of magnitude through a developed annealing process. The promotion of near-room-temperature performance with the doubled carrier mobility is also verified by the round-robin test. This manuscript is well-written, and the arguments are sound and thus deserve to be published in Nature Communications.

Nevertheless, there are some potentials for improvements of the manuscript, which will enhance the impact of the paper in the future (as listed below):

1. The scale bar in Fig. 1b-annealed one is missing.

2. I would suggest separating a few important equations into individual lines.

3. Indeed, improved room temperature mobility can be expected with the increase of grain size, as observed in NbFeSb reported by Ran et al. For NbFeSb, the enlarged grain size is obtained by high-temperature hot-pressing. I believe the alloying effect of Ta makes the grain growth process more complicated. Can you further promote the grain size of half-Heusler by hot-pressing and annealing at high temperatures together? Please compare the growth mechanism and add the discussion in this work.

4. Will the room temperature performance be further improved in the single crystal sample? What is the limit of the room temperature ZT for half-Heusler, a material system with such a high lattice thermal conductivity?

5. The superior cooling density in this work benefits from the high power factor of NbFeSb-based materials. In previous Nb_{0.95}Ti_{0.05}FeSb work (doi/10.1073/pnas.1617663113), an even higher power factor of ~100 μW/cm-K² was reported. Do the authors expect this

material could provide higher cooling density?

Reviewer #3 (Remarks to the Author):

Comments to the authors

1. Another thermoelectric material exhibited a high power factor at room temperature with reasonable zT (<https://doi.org/10.1073/pnas.1617663113>; $\text{Nb}_{0.95}\text{Ti}_{0.05}\text{FeSb}$ $\sim 106 \mu\text{W cm}^{-1}\text{K}^{-2}$). How is the current material compared with the existing $\text{Nb}_{0.95}\text{Ti}_{0.05}\text{FeSb}$ half-Heusler material?

2. Generally, grain growth happens during the long hours of annealing. However, a driving force should be required for this grain boundary annihilation/movement (grain boundary area/ chemical potential difference across the boundaries). Also, the solute drag effect of the boundaries certainly restricts grain growth. How is the Sb-pressure control developed in the $\text{Nb}_{0.55}\text{Ta}_{0.40}\text{Ti}_{0.05}\text{FeSb}$ compound or alloy to eradicate the boundaries and turn to larger grains? Hence, the authors need to explain the procedure/methodology for obtaining the larger grains under the influence of Sb-pressure control concerning activation energy and temperature.

3. The authors stated that the zT of the hot press samples is less than the annealed sample. This may be due to the annealing process that reduces the defects and increases the grain growth, which was resulted in a high power factor with un-changed lattice thermal conductivity (κ_l). However, the λ (mean free path) is proportional to lattice thermal conductivity for larger grains should exhibit higher κ_l ? A suitable explanation must be incorporated regarding the lower lattice thermal conductivity as it influences the quality factor, which is directly proportional zT .

4. All the variables (T_c , κ , PF, z , ΔT etc.) used in Equation 1 need to be tabulated for the current $\text{Nb}_{0.55}\text{Ta}_{0.40}\text{Ti}_{0.05}\text{FeSb}$ half-Heusler material along with the reference material (Bismuth Telluride). This gives a clear idea about the enhancement in $Q_{(c_max)}$ with respect to temperature and the significance of the dependent variables.

5. How is the “COP typically lower than its maximum value at low cooling density”, and this statement should be referenced? As the leg length decreases to 0.2 mm (micro) from 2 mm (conventional), the $Q_{(c_max)}$ increases to 210% (Fig. 4(f)). Authors should clarify how the PF value is considered for the hH-YbAl₃ module in the estimation of $Q_{(c_max)}$ employing equation 1 rather single leg.

6. In page 8, the statement “Interestingly, κ_L did not increase with grain size due to the weak phonon scattering from the grain boundary” should be more clearly explained. As the grain boundary density is reduced with larger grains from the annealing, how is the κ_L lowered with weak phonon scattering?

7. In the “High cooling capacity half-Heusler module performance” section, the leg lengths are ~ 0.2 mm and 0.3 mm (mm or cm). It is too small, and the authors are advised to check the same with the standard thermoelectric device leg length at Kelk and Thermion, respectively. As these values greatly influence the $Q_{(c_max)}$ calculations.

8. A suitable reference needs to be added for “the maximum cooling power density of the module ($Q_{(c_max)}$)” equation.

9. In-Page 5, the statement is included as “excellent conversion efficiency of 12% at ΔT of

600 K. The same is not reflected in Fig. 1 (c), as mentioned in the text. Authors must provide the necessary power efficiency estimation data in the supplementary information. Also, describe the contact resistance effect on the output power efficiency if you have considered it.

10. Fig. S3 (κ) claims that the Sb-rich region from the grain boundary is eliminated. The segregated Sb is dissolved in the matrix after 2 days of annealing at 1143 K under the Sb pressure control. What is actually Sb pressure controlled? Does only Sb is regulated through an annealing process?

11. In Fig.5 (d), the authors provided the power efficiency data for p-Nb_{0.55}Ta_{0.40}Ti_{0.05}FeSb and n-Zr_{0.44}Hf_{0.44}Ti_{0.12}NiSn_{0.9}Sb_{0.01} uni-couple, which is 40% higher than the previous work (Ref. 46). However, the contact resistance plays a vital role in the final device efficiency. Further, the authors did not provide any information about the uni-couple's contact engineering/contact resistance/total device resistance; the same need to be incorporated.

12. The COP formulae should be placed in the main/supplementary manuscript to explain the dependent and independent parameters for a better COP. Also, a detailed explanation needs to be added for the COP change concerning zT value and ΔT of the hH-YbAl₃ device.

13. Authors should provide the contact resistance plots (Fig. 5(a)) for Nb_{0.55}Ta_{0.40}Ti_{0.05}FeSb and Zr_{0.44}Hf_{0.44}Ti_{0.12}NiSn_{0.9}Sb_{0.01} individual or device.

14. The authors claimed that “The maximum $Q_{(c_max)}$ of ~ 88 W cm⁻² can be achieved for hH-YbAl₃ device when the length of TE leg is projected to be 0.2 mm (Fig. 4f)” which is 210% higher than existing Bi₂Te₃ TE devices, under the identical conditions of $0.2 \mu\Omega/\text{cm}^2$, thermal conductance: 0.5 mm²K/W, fill factor: 60% and these values calculated by changing the leg height. However, these are theoretical predictions, and obtaining such a low contact resistance is a big challenge in actual devices. Hence, the authors should mention a few probable candidate contact materials which can possess low specific contact resistance. Is there any data available with the authors for the said leg length (0.2 mm)? If yes, the same may be incorporated into the current manuscript.

15. The authors claimed that “excellent conversion efficiency of 12% at ΔT of 600 K for p-Nb_{0.55}Ta_{0.40}Ti_{0.05}FeSb and n-Zr_{0.44}Hf_{0.44}Ti_{0.12}NiSn_{0.9}Sb_{0.01} uni-couple”. The contact resistance will vary at the interface (metal contact-leg) with the temperature. Hence authors should provide the contact resistance experimental data for 600 K.

Dear Editor and Reviewers,

At the outset, we would like to thank you for providing us constructive feedback on our manuscript. The comments have been very helpful in further improving the quality of manuscript. We have revised our manuscript based on the reviewers' comments and questions. Hope you will find the changes to the manuscript satisfactory.

Below we provide point-by-point response to reviewer comments:

Reviewer #1 (Remarks to the Author):

In this work, the authors fabricated HH materials and devices and then investigated the cooling and power generation efficiencies. They want to demonstrate that HH alloys are more suitable for cooling applications than bismuth telluride alloys by emphasizing that HH alloys have higher power factors, leading to higher cooling power density. Although the present work may be technically important, I am quite hesitating to recommend it to publication in Nature Communications because of the following concerns.

1. The hH alloys have been well studied and the compositions focused in this work show very common ZT values.

Reply: In practical applications, the **average ZT** is more important than the peak ZT of thermoelectric materials, because the performances of thermoelectric devices, such as the conversion efficiency and maximum cooling temperature, are both determined by the average ZT instead of peak ZT. Particularly in this work, we are targeting on the cooling performance using hH materials, which requires the high power factor and ZT near temperature. In order to pursue the limit of high average ZT of hH, we carefully designed the composition and microstructure of (Nb,Ta)FeSb alloy and developed powder metallurgy and heat-treatment process to reduce the density of grain boundaries and defects. As compared to state-of-the-art results, a **significant improvement of ZT near room temperature and record high average ZT of 0.86** in the temperature range of 300-873 K was obtained for half-Heusler alloys (Fig. 4f). In addition, it is noteworthy that the high performance was verified by the round robin test at Oak Ridge National Laboratory.

2. Most importantly, for cooling applications, the maximum temperature difference is another important parameter. For commercialized bismuth telluride modules, it can reach 70 degree C, but how about the present hH module?

Reply: We agree with reviewer about the importance of maximum temperature difference in thermoelectric cooling devices. It is worth to note that **both the maximum temperature difference under zero heat load and the maximum cooling density under zero temperature difference are the most important parameters to evaluate the performance of device**. Particularly, in this study, we focused on the application of thermal management for high-power electronics, which requires the maximum cooling density under small ΔT instead of ΔT maximization. The module of hH-YbAl₃ demonstrated in this work verifies the idea that the power factor is critical, especially when the ΔT is small. In the high-density cooling application, Nb_{0.55}Ta_{0.40}Ti_{0.05}FeSb material developed in this work has advantage over Bi₂Te₃ under small ΔT (Fig. 5). Practically, the hH-YbAl₃ device with given ΔT shows similar maximum COP as compared with Bi₂Te₃-based device. More importantly, the maximum cooling density of hH-YbAl₃ module is ~210% higher than that of commercial Bi₂Te₃-based

modules, which is our primary focus (Fig. 5e).

For reviewer's information, our uncouple hH-Bi₂Te₃ device shows a ΔT of ~ 55 K, which is slight lower than the predicted value of ~ 62 K due to the unoptimized contact condition at the interfaces (Fig. R1). For hH-YbAl₃ uncouple device, a relatively low ΔT of ~ 26 K is obtained due to the low ZT value of YbAl₃ (Fig. R1).

Fig. R1. The temperature dependent maximum cooling density of (a) hH-Bi₂Te₃ device and (b) hH-YbAl₃ device.

- The authors made comparison between hH and Bi₂Te₃, but no detailed information can be found. The authors cited a reference from the same author group, but also no materials compositions and properties are reported in this paper.

Reply: All the details about synthesis method, composition and thermoelectric properties of the materials can be found in Supplementary Information. The synthesis method of Bi₂Te₃ materials can be found in “Synthesis of Bi₂Te₃ based materials. The thermoelectric properties of p-type Sb_{1.67}Bi_{0.33}Te₃ and n-type Bi₂Te_{2.7}Se_{0.3} materials are shown in Fig. S10. The thermoelectric properties of n-type Zr_{0.44}Hf_{0.44}Ti_{0.12}NiSn_{0.9}Sb_{0.01} materials are shown in Fig. S12. We also added the description of materials preparation in Supplementary Information accordingly.

Please find the added content in Supplementary Information as below:

The synthesis and properties of n-type Zr_{0.44}Hf_{0.44}Ti_{0.12}NiSn_{0.9}Sb_{0.01} hH composition.

The synthesized process of Zr_{0.44}Hf_{0.44}Ti_{0.12}NiSn_{0.9}Sb_{0.01} alloys is similar with our previous work.¹¹ Stoichiometric amounts of high purity metal precursors of hafnium piece (99.9%, Alfa Aesar), zirconium slug (99.9%, Alfa Aesar), titanium wire (99.9%, Alfa Aesar), nickel slug (99.995%, Alfa Aesar), antimony shot (99.999%, Alfa Aesar), and tin wire (99.95%, Alfa Aesar) were mixed and melted by radiofrequency induction melting under an argon atmosphere for 5 min. The ingots were rotated and remelted several times to ensure homogeneity. The resulting ingots were pulverized and transferred in a stainless steel container with grinding balls under an argon environment in a glove box. High energy ball milling was conducted for 4 hours using SPEX mixer/mill (Model 8000D, SPEX SamplePrep, Metuchen, NJ). The mixed powders were consolidated by Spark Plasma Sintering (SPS, Model Dr. Sinter-625V, Fuji, Japan) at 1150 °C under a pressure of 80 MPa for 5 min, yielding fully dense pellets.

Fig. S12. The thermoelectric transport properties of $\text{Zr}_{0.44}\text{Hf}_{0.44}\text{Ti}_{0.12}\text{NiSn}_{0.9}\text{Sb}_{0.01}$ sample prepared in this work. **a**, The temperature dependent electrical conductivity (σ) and Seebeck coefficient (S). **b**, PF . **c**, total thermal conductivity (κ_{tot}). **d**, zT value.

The synthesis and properties of $(\text{Bi, Sb})_2(\text{Te, Se})_3$ -based materials.

The synthesized process of $(\text{Bi, Sb})_2(\text{Te, Se})_3$ -based materials alloys is similar with our previous work.⁷ High-purity elements of Bi (99.9%, pieces), Sb (99.9%, shots), Te (99.9%, lumps), Se (99.9%, pellets) weighted stoichiometrically, sealed in a sealed quartz tube and melted for 24 hours at 1000 °C inside a conventional furnace following by quenching. The ingot was then loaded into a hardened steel vial along with steel balls with ball to powder ratio of 1:1 and ball milled for 3 hours using SPEX mixer/mill (Model 8000D, SPEX SamplePrep) to obtain homogeneous powder. The powders were then consolidated in a cylindrical graphite die using spark plasma sintering (SPS) at 520 °C (for p-type) and 450 °C (for n-type) under constant 30 MPa pressure for 2 minutes.

Fig. S10. The thermoelectric transport properties of $(\text{Bi, Sb})_2(\text{Te, Se})_3$ -based samples used in this work. **a**, The temperature dependent electrical conductivity (σ). **b**, Seebeck coefficient (S). **c**, Power

factor ($S^2\sigma$). **d**, Total thermal conductivity (κ_{tot}). **e**, Lattice thermal conductivity (κ_L). **f**, zT value. The BiTe materials are p-type $\text{Bi}_{0.33}\text{Sb}_{1.67}\text{Te}_3$ and n-type $\text{Bi}_2\text{Te}_{2.7}\text{Se}_{0.3}$.

4. For the power generation, the efficiency is not high for such a large temperature difference. What are the electrode and contact layer used for the power generation devices?

Reply: Cu was used as electrode. As the major target of this work is cooling application, we used liquid metal as the contact layer and thermal interface material in this work, which provides excellent electrical and thermal contact. The details can be found in Supplementary Information — Materials and Methods section.

As the high temperature thermoelectric materials, the average ZT of hH is difficult to be improved, which results in few reported efficiencies of **single stage hH** devices reaching the value of 10 (*Joule* 2020, 4, 2475). Please find the comparison of conversion efficiency for state-of-the-art **single stage hH** devices reported so far in Table R1 (most of them with ΔT of 650~700 K). The conversion efficiency of ~12% obtained in this work is higher than all the other reported **single stage hH device**, especially considering of smaller ΔT of 600 K among these hH devices.

Table R1 The state-of-the-art conversion efficiency of single stage hH device

Literature	Conversion efficiency	ΔT
This study	12%	600 K
Mater. Today 2020 , 36, 63	10.7%	674 K
Energy Environ. Sci. , 2019 , 12, 3390	9.6%	665 K
Adv. Energy Mater. 2020 , 10, 2000888	8.3%	655 K
J. Mater. Res. 2011 , 26, 2795	8.8%	700 K
Joule 2020 , 4, 2475	10.5%	680 K

5. The authors spent a lot of spaces to emphasize the importance of grain growth, but it is also well known and the method has been reported.

Reply: The grain growth of materials is mostly achieved by high temperature thermal treatment, e.g., under vacuum. The hH alloys has very high melting point up to 1273 K and excellent thermal stability, which makes the growth of grain in hH alloys difficult and has been rarely reported. As shown in Fig. S2, no obvious grain growth can be observed for as hot-pressed $\text{Nb}_{0.55}\text{Ta}_{0.40}\text{Ti}_{0.05}\text{FeSb}$ samples after conventional annealing process (at 1123 K under vacuum for more than 6 days). In this study, **a Sb pressure controlled annealing process has been developed**. The pinning behavior of Sb-rich phase in grain boundary was discovered leading to the abnormal growth of grain by 3 orders of magnitude by removing the Sb-rich phase. The mechanism and method behind this work is different with conventional process. As discussed in manuscript, the enlarged the gain size largely reduces scattering from the grain boundaries leading to the significantly enhanced mobility and record high average ZT among the hH alloys.

We further analyzed and calculated the kinetics of grain growth as a function of annealing time under

different temperatures to explain the Sb pressure-controlled annealing process and have added the corresponding discussion in manuscript as below:

Fig. 3 shows the estimated grain growth kinetics and grain size as a function of annealing time under different temperatures for $\text{Nb}_{0.55}\text{Ta}_{0.40}\text{Ti}_{0.05}\text{FeSb}$ sample under Sb pressure-controlled annealing condition. As the grain growth are thermally activated, the rate of boundary motion is determined by annealing temperature (T), the activation energy of boundary motion (Q_{gb}), and driving force of grain growth (p). According to the Turnbull expression, the grain-boundary velocity (x) can be given as⁴¹:

$$x = nmp = nm_0 \exp\left(-\frac{Q_{gb}}{k_B T}\right) p \quad [2]$$

where n is the normal of grain-boundary segment, and m and m_0 are the actual and initial mobility of grain boundary, respectively. The Q_{gb} can be reduced by eliminating any segregations at grain boundaries, e.g., removing the Sb-rich phase in this study.

The p is the summation of acting forces for boundary migration:

$$p = p_{gb} - p_p - p_s \quad [3]$$

The p_{gb} is the driving force determined by the surface energy of grain boundaries (γ) with average diameter of grains (D)⁴², which is the major source of p :

$$p_{gb} = 2\gamma/D \quad [4]$$

The p_p is the retarding force due to the pinning force of the particles at grain boundaries determined by volume fraction (V_p) and diameter of the particles (r). The elimination of Sb-rich phase at 1143 K can significantly reduce the V_p so that the p_p is reduced.

$$p_p = 2V_p/r \quad [5]$$

The p_s is the retarding force from solute drag effect determined by the interaction energy (E) with boundaries and composition profile (C)⁴³:

$$p_s = N_V \int_{-\infty}^{+\infty} (C - C_0) \frac{dE}{dx} dx \quad [6]$$

where N_V is the number of solute atoms per unit volume. In conventional annealing process, the solute atoms tend to segregate at the grain boundaries and thereby retard the migration of grain boundaries. However, during the Sb pressure-controlled annealing process, the elimination of Sb-rich phase also contributes to the reduction of p_s . Overall, the grain growth of sample can be promoted significantly. The estimated m is 0.005, 0.127 and 4.700 $\text{cm}^2 \text{V}^{-1} \text{s}$, at 1073 K, 1113 K and 1143 K, respectively. And the calculated activation energy Q_{gb} is ~ 8.2 eV for the hH alloys annealed with Sb-pressure control. Consequently, the estimated grain size of samples annealed at 1073 K, 1113 K and 1143 K for 2 days increased to ~ 5 μm , 25 μm and 150 μm , respectively, which is generally consistent with the experimental results (Fig. 2).

Fig. 1 The grain growth kinetics and grain size as a function of annealing time under different temperatures for $\text{Nb}_{0.55}\text{Ta}_{0.40}\text{Ti}_{0.05}\text{FeSb}$ sample under Sb pressure-controlled condition. **a**, Driving force and velocity of grain growth. **b**, Grain size.

6. About Figure 2 e to h, it is written as “The vacancies in crystalline lattice, where the internal energy is accumulated during ball milling and hot-pressing process, were minimized after this process”. Very unclear, more explanations are needed.

Reply: We appreciate the suggestions and have revised it accordingly as below:

The vacancy in the lattice of the samples where the internal energy is stored during ball-milling and hot-pressed process were also minimized after this process, as confirmed in SEM and TEM results (Fig. 2e-h and Fig. S3k-j)

7. In the abstract, the second sentence seems uncompleted.

Reply: Thanks for the comment and we have revised it accordingly as below.

In thermoelectric (TE) cooling, high cooling power density, which is correlated directly to the power factor of TE materials, is imperative for hot-spot and high-power electronics thermal management, while the state-of-the-art Bi_2Te_3 -based TE materials shows limited cooling power density so far.

8. In the conclusion, the cooling power density is written as “1-3.5 W cm⁻²”, much lower than that in the text, why?

Reply: The power density of 1-3.5 W cm⁻² is for the hH-YbAl₃ device with leg length of 7.7 mm that

has been tested experimentally. If the leg length is projected to 0.2 mm, which is used in the most advanced commercial micro-TEC, the maximum power density of $\sim 88 \text{ W cm}^{-2}$ will be achieved by hH-YbAl₃ device as shown in Fig. 5f. We have modified the discussion in the manuscript to avoid any misunderstanding as below:

The measured $Q_{c_{max}}$ of the hH-YbAl₃ device at ΔT of 5 K is found to be 3.6 W cm^{-2} (with leg length of 7.7 mm), which is twice that of the Bi₂Te₃-based device, 1.7 W cm^{-2} , with same leg length and ΔT (Fig. 4b,c and Fig. S9).

The maximum $Q_{c_{max}}$ of $\sim 88 \text{ W cm}^{-2}$ can be achieved for hH-YbAl₃ device when the length of TE leg is projected to be 0.2 mm (Fig. 4f), which is $\sim 210\%$ higher than that of commercial Bi₂Te₃-based modules.

Reviewer #2 (Remarks to the Author):

The publication on high power cooling performance of (Nb,Ta)FeSb is an interesting contribution to half-Heusler thermoelectric. The grain size of the half-Heusler materials is increased by orders of magnitude through a developed annealing process. The promotion of near-room-temperature performance with the doubled carrier mobility is also verified by the round-robin test. This manuscript is well-written, and the arguments are sound and thus deserve to be published in Nature Communications.

Nevertheless, there are some potentials for improvements of the manuscript, which will enhance the impact of the paper in the future (as listed below):

1. The scale bar in Fig. 1b-annealed one is missing.

Reply: Thanks for the comments. We have added the scale bar.

2. I would suggest separating a few important equations into individual lines.

Reply: Thanks for the comments. We have modified the equations accordingly.

3. Indeed, improved room temperature mobility can be expected with the increase of grain size, as observed in NbFeSb reported by Ran et al. For NbFeSb, the enlarged grain size is obtained by high-temperature hot-pressing. I believe the alloying effect of Ta makes the grain growth process more complicated. Can you further promote the grain size of half-Heusler by hot-pressing and annealing at high temperatures together? Please compare the growth mechanism and add the discussion in this work.

Reply: Owing to the high thermal stability, NbFeSb sample can be hot pressed at very high temperature up to 1373K. The promoted atomic diffusivity at elevated temperature enables the abnormal grain growth within the short hot-pressing time of 5 mins. It is very interesting idea to find out whether our annealing process can further remarkably improve the grain size of the as-hot-pressed NbFeSb large-grain sample. We will verify this idea in our future work.

For the Ta-alloyed sample, the phase transition temperature of $\text{Nb}_{0.55}\text{Ta}_{0.40}\text{Ti}_{0.05}\text{FeSb}$ decreases to 1173 K. The hot-pressing of $\text{Nb}_{0.55}\text{Ta}_{0.40}\text{Ti}_{0.05}\text{FeSb}$ can only be conducted at 1073 K which is 300 K lower than NbFeSb. Meanwhile, the solute atom of Ta changes the activation energy of the grain boundary via solute drag mechanism, has remarkable influence on the kinetics of grain growth process. Therefore, owing to the limited hot-pressing temperature and solute drag effect, no remarkable grain growth was observed during the hot-pressing process of $\text{Nb}_{0.55}\text{Ta}_{0.40}\text{Ti}_{0.05}\text{FeSb}$. The grain size of the as-hot-pressed sample can only be further improved by two days annealing with controlled Sb pressure at a lower temperature of 1143 K.

We have added the above discussions in the revision.

Because the decrease of phase transition temperature of $\text{Nb}_{0.55}\text{Ta}_{0.40}\text{Ti}_{0.05}\text{FeSb}$ compound after Ta-alloying to 1173 K, the hot-pressing temperature of $\text{Nb}_{0.55}\text{Ta}_{0.40}\text{Ti}_{0.05}\text{FeSb}$ sample can only be

conducted at 1073 K, which is about 300 K lower than that of NbFeSb compound. Thereby, no remarkable grain growth was observed during the hot-pressing process of Nb_{0.55}Ta_{0.40}Ti_{0.05}FeSb sample under vacuum condition at such a limited hot-pressing temperature.

4. Will the room temperature performance be further improved in the single crystal sample? What is the limit of the room temperature ZT for half-Heusler, a material system with such a high lattice thermal conductivity?

Reply: The carrier mobility in material is mainly determined by various scattering sources, which will be largely eliminated in high-quality single crystals. In the single crystal sample, the power factor will be further improved so that better room temperature performance can be expected. However, for the state-of-art half-Heusler, the power factor of Nb_{0.55}Ta_{0.40}Ti_{0.05}FeSb is already one of the best values. We have also attempted to anneal the Nb_{0.95}Ti_{0.05}FeSb sample to improve the performance by a conventional annealing process. However, due to the high intrinsic thermal conductivity, no further enhancement has been observed till now.

Half-Heusler system consists of hundreds of materials, including a variety of thermoelectric candidates with promising properties that have not been thoroughly studied. In our previous works (*J. Mater. Chem. A*, 2020, 8, 4790), the AgCu(Te,Se) with high room temperature ZT of 0.74 was discovered. The soft phonon mode leads to the ultra-low lattice thermal conductivity ($<0.4 \text{ W m}^{-1} \text{ K}^{-1}$) near the phase transition temperature of 390 K which makes AgCu(Te,Se) to be an outstanding room temperature thermoelectric material different from the conventional half-Heuslers. We believe that due to its great potential, the discovery of half-Heusler with room temperature performance competitive to Bi₂Te₃ system can be reasonable expected.

5. The superior cooling density in this work benefits from the high power factor of NbFeSb-based materials. In previous Nb_{0.95}Ti_{0.05}FeSb work (doi/10.1073/pnas.1617663113), an even higher power factor of $\sim 100 \mu\text{W}/\text{cm}\cdot\text{K}^2$ was reported. Do the authors expect this material could provide higher cooling density?

Reply: We appreciate the suggestions. As compared to Nb_{0.95}Ti_{0.05}FeSb material, Ta alloying can significantly improve the μ/κ_L ratio resulting in $\sim 60\%$ enhancement of average zT value in the temperature range of 300-900 K, as shown in Fig. R2. Ta substitution can introduce mass fluctuation without lattice stress owing to the small difference in lattice constant of less than 1% between NbFeSb and TaFeSb, which largely preserves the high mobility and high power factor realized in Nb_{0.95}Ti_{0.05}FeSb material. The Nb_{0.55}Ta_{0.40}Ti_{0.05}FeSb shows a comparable power factor near room temperature ($\sim 80 \mu\text{W}\cdot\text{cm}^{-1}\cdot\text{K}^{-2}$) with that of Nb_{0.95}Ti_{0.05}FeSb material ($\sim 100 \mu\text{W}\cdot\text{cm}^{-1}\cdot\text{K}^{-2}$). More importantly, it shows much higher ZT near room temperature and average ZT that will lead to the enhancement of maximum cooling density under small ΔT .

Fig. R2 The power factor and ZT of $\text{Nb}_{0.55}\text{Ta}_{0.40}\text{Ti}_{0.05}\text{FeSb}$ sample and $\text{Nb}_{0.95}\text{Ti}_{0.05}\text{FeSb}$ sample reported in the literature.

Reviewer #3 (Remarks to the Author):

Comments to the authors

1. Another thermoelectric material exhibited a high power factor at room temperature with reasonable zT (<https://doi.org/10.1073/pnas.1617663113>; $\text{Nb}_{0.95}\text{Ti}_{0.05}\text{FeSb}$ $\sim 106 \mu\text{W cm}^{-1} \text{K}^{-2}$). How is the current material compared with the existing $\text{Nb}_{0.95}\text{Ti}_{0.05}\text{FeSb}$ half-Heusler material?

Reply: We appreciate the suggestions. As compared to $\text{Nb}_{0.95}\text{Ti}_{0.05}\text{FeSb}$ material, Ta alloying can significantly improve the μ/κ_L ratio resulting in $\sim 60\%$ enhancement of average zT value in the temperature range of 300-900 K, as shown in Fig. R2. Ta substitution can introduce mass fluctuation without lattice stress owing to the small difference in lattice constant of less than 1% between NbFeSb and TaFeSb , which largely preserves the high mobility and high power factor realized in $\text{Nb}_{0.95}\text{Ti}_{0.05}\text{FeSb}$ material. The $\text{Nb}_{0.55}\text{Ta}_{0.40}\text{Ti}_{0.05}\text{FeSb}$ shows a comparable power factor near room temperature ($\sim 80 \mu\text{W}\cdot\text{cm}^{-1}\cdot\text{K}^{-2}$) with that of $\text{Nb}_{0.95}\text{Ti}_{0.05}\text{FeSb}$ material ($\sim 100 \mu\text{W}\cdot\text{cm}^{-1}\cdot\text{K}^{-2}$). More importantly, it shows much higher ZT near room temperature and average ZT that will lead to the enhancement of maximum cooling density under small ΔT .

Fig. R2 The power factor and ZT of $\text{Nb}_{0.55}\text{Ta}_{0.40}\text{Ti}_{0.05}\text{FeSb}$ sample and $\text{Nb}_{0.95}\text{Ti}_{0.05}\text{FeSb}$ sample reported in the literature.

2. Generally, grain growth happens during the long hours of annealing. However, a driving force should be required for this grain boundary annihilation/movement (grain boundary area/ chemical potential difference across the boundaries). Also, the solute drag effect of the boundaries certainly restricts grain growth. How is the Sb-pressure control developed in the $\text{Nb}_{0.55}\text{Ta}_{0.40}\text{Ti}_{0.05}\text{FeSb}$ compound or alloy to eradicate the boundaries and turn to larger grains? Hence, the authors need to explain the procedure/methodology for obtaining the larger grains under the influence of Sb-pressure control concerning activation energy and temperature.

Reply: Thanks for the suggestions. In general, we found that grain boundary pinning behavior of Sb-rich phase imparts dynamic resistance to grain growth through TEM investigation (Fig. S3a-h). An Sb pressure-controlled annealing process is developed to remove the excess Sb and provide an Sb deficient environment so that the pinning effect can be eliminated. As a result, no Sb-rich phase was observed at grain boundaries after Sb pressure-controlled annealing at 1143 K for 2 days (Fig. S3i-j), and grain sizes were significantly increased by 3 orders of magnitude, from $\sim 200 \text{ nm}$ to $\sim 150 \mu\text{m}$, due

to the substantially improved grain boundary migration rate (Fig. 2a-d).

As the reviewer suggested, we calculated the grain growth and driving force as a function of annealing temperature to explain the Sb pressure-controlled annealing process and added the corresponding discussion in manuscript as below:

Fig. 3 shows the estimated grain growth kinetics and grain size as a function of annealing time under different temperatures for $\text{Nb}_{0.55}\text{Ta}_{0.40}\text{Ti}_{0.05}\text{FeSb}$ sample under Sb pressure-controlled condition. As the grain growth are thermally activated, the rate of boundary motion is determined by annealing temperature (T), the activation energy of boundary motion (Q_{gb}), and driving force of grain growth (p). According to the Turnbull expression, the grain-boundary velocity (x) can be given as⁴⁰:

$$x = nmp = nm_0 \exp\left(-\frac{Q_{\text{gb}}}{k_{\text{B}}T}\right) p \quad [2]$$

where n is the normal of grain-boundary segment, and m and m_0 are the actual and initial mobility of grain boundary, respectively. The Q_{gb} can be reduced by eliminating any segregations at grain boundaries, e.g., removing the Sb-rich phase in this study.

The p is the summation of acting forces for boundary migration:

$$p = p_{\text{gb}} - p_{\text{p}} - p_{\text{s}} \quad [3]$$

The p_{gb} is the driving force determined by the surface energy of grain boundaries (γ) with average diameter of grains (D)⁴¹, which is the major source of p :

$$p_{\text{gb}} = 2\gamma/D \quad [4]$$

The p_{p} is the retarding force due to the pinning force of the particles at grain boundaries determined by volume fraction (V_{p}) and diameter of the particles (r). The elimination of Sb-rich phase at 1143 K can significantly reduce the V_{p} so that the p_{p} is reduced.

$$p_{\text{p}} = 2V_{\text{p}}/r \quad [5]$$

The p_{s} is the retarding force from solute drag effect determined by the interaction energy (E) with boundaries and composition profile (C)⁴²:

$$p_{\text{s}} = N_{\text{V}} \int_{-\infty}^{+\infty} (C - C_0) \frac{dE}{dx} dx \quad [6]$$

where N_{V} is the number of solute atoms per unit volume. In conventional annealing process, the solute atoms tend to segregate at the grain boundaries and thereby retard the migration of grain boundaries. However, during the Sb pressure-controlled annealing process, the elimination of Sb-rich phase also contributes to the reduction of p_{s} . Overall, the grain growth of sample can be promoted significantly. The estimated m is 0.005, 0.127 and 4.700 $\text{cm}^2 \text{V}^{-1} \text{s}$, at 1073 K, 1113 K and 1143 K, respectively. And the calculated activation energy Q_{gb} is ~ 8.2 eV for the hH alloys annealed with Sb-pressure control. Consequently, the estimated grain size of samples annealed at 1073 K, 1113 K and 1143 K for 2 days increased to ~ 5 μm , 25 μm and 150 μm , respectively, which is consistent with the experimental results (Fig. 2).

Fig. 2 The grain growth kinetics and grain size as a function of annealing time under different temperatures for $\text{Nb}_{0.55}\text{Ta}_{0.40}\text{Ti}_{0.05}\text{FeSb}$ sample under Sb pressure-controlled condition. **a**, Driving force and velocity of grain growth. **b**, Grain size.

3. The authors stated that the zT of the hot press samples is less than the annealed sample. This may be due to the annealing process that reduces the defects and increases the grain growth, which was resulted in a high power factor with un-changed lattice thermal conductivity (κ_l). However, the λ (mean free path) is proportional to lattice thermal conductivity for larger grains should exhibit higher κ_l ? A suitable explanation must be incorporated regarding the lower lattice thermal conductivity as it influences the quality factor, which is directly proportional zT .

Reply: The phonon mean free path in half-Heusler is lower than 100 nm (*Nat Commun*, 2018. 9, 1721). The grain size of sample studied in this work is in the range of ~ 200 nm to ~ 150 μm , which is larger than the mean free path of phonon. Therefore, the grain boundaries in this work have limited impact on the phonon propagation and the lattice thermal conductivity. Similar lattice thermal conductivity is also reported in NbFeSb sample with grain size range from 200 nm to 1.5 μm , as the author mentioned in their work “We also observe that the lattice thermal conductivity hardly changes within the range of grain sizes studied in this work.” (*PANS*, 2016. 113, 13576-13581) Similar results are also observed in $\text{Mg}_3(\text{Bi,Sb})_2$ -based materials (*Adv Mater*, 2019, 31, 1902337).

The mean free path of electron is even lower than the phonon, and the influence of grain boundaries on the carrier mobility is mainly attributed to the potential barrier in hH and $\text{Mg}_3(\text{Bi,Sb})_2$ -based materials (*Energy Environ Sci*, 2018, 11, 429-434 and *Energy Environ Sci*, 2022, 15, 1406-1422).

We have added corresponding discussions and literature in manuscript as below:

The mean free path (λ) of phonon in hH is lower than 100 nm³¹, which is less than the grain size of hH

in this study. Consequently, κ_L did not increase with the increase of grain size and reduction of grain boundaries due to the weak phonon scattering from the grain boundary¹⁵, which has also been observed in $Mg_3(Bi,Sb)_2$ system^{14,43}.

4. All the variables (T_c , κ , PF , z , ΔT etc.) used in Equation 1 need to be tabulated for the current $Nb_{0.55}Ta_{0.40}Ti_{0.05}FeSb$ half-Heusler material along with the reference material (Bismuth Telluride). This gives a clear idea about the enhancement in $Q_{(c_max)}$ with respect to temperature and the significance of the dependent variables.

Reply: We appreciate the suggestions and have added the Table. S1 in the supporting information section as below:

Table S 1 The parameters of p-type hH and BiTe materials near room temperature used in this study for high cooling density ($Q_{c_{max}}$) consideration.

Parameters	p-type $Nb_{0.55}Ta_{0.40}Ti_{0.05}FeSb$	p-type $Bi_{0.33}Sb_{1.67}Te_3$
T_c (K)	298	298
ΔT (K)	5	5
z (K^{-1})	1.25e-3	3.6e-3
PF ($\mu W \cdot cm^{-1} \cdot K^{-2}$)	78	40
κ ($W \cdot m^{-1} \cdot K^{-1}$)	6.24	1.09
l (mm)	7.7	7.7

5. How is the “COP typically lower than its maximum value at low cooling density”, and this statement should be referenced? As the leg length decreases to 0.2 mm (micro) from 2 mm (conventional), the $Q_{(c_max)}$ increases to 210% (Fig. 4(f)). Authors should clarify how the PF value is considered for the hH- $YbAl_3$ module in the estimation of $Q_{(c_max)}$ employing equation 1 rather single leg.

Reply: Equation 1 is presented to explain the relation between the device cooling performance and thermoelectric properties, mostly for a quick assessment of single leg. As the thermoelectric device is composed of both n- and p-type materials and thermoelectric properties is temperature dependent, numerical analysis is conducted to give reliable prediction. Please find details in “Numerical analysis of TE device” in supplementary information.

Theoretically, high COP and high cooling density are two terminals of cooling device, as shown in Fig. S9c,d, which means the maximum COP and the maximum cooling density cannot be achieved simultaneously. For instance, if the cooling density is plotted as a function of COP (Fig. S9d), we can clearly observe that the COP reaches its peak value with low cooling density, and then decreases with the increase of cooling density. Thus, under a high heat load (maximum cooling density point), the COP decreases to ~ 0.5 . Similar results can also be find in the cooling performance based on $Mg_3(Bi,Sb)_2$ materials (Joule, 2022. 6, 193).

We have modified the figures and added corresponding discussion in manuscript and Supplementary Information as below:

Manuscript: The COP of the hH- $YbAl_3$ device, which is determined by the zT value, ΔT and current

I , is demonstrated in Fig. 5d. The current applied to the device determines heat load, as shown in Fig. 5b. Practically, when TE cooling devices work under a high heat load, the COP is usually lower than its maximum value at low cooling density (Fig. S9c,d)⁴⁴.

Supplementary Information: As shown in Fig. S9c,d, high COP and high cooling density are two terminals of cooling device, which means maximum COP and cooling density cannot be achieved simultaneously. We can clearly observe that the COP reaches its peak value with low cooling density, and then decreases with the increase of cooling density (Fig. S9d). For instance, under a high heat load ($Q_{c,max}$ of $\sim 4 \text{ W}\cdot\text{cm}^{-2}$), the COP decreases to ~ 0.5 for both YbAl₃-hH measured device and hypothetical hH device; while, under a low heat load ($Q_{c,max}$ of ~ 0.75 and $0.5 \text{ W}\cdot\text{cm}^{-2}$ for hH-YbAl₃ measured device and hypothetical hH device, respectively), the COP increases to ~ 2.5 and 4 for hH-YbAl₃ measured device and hypothetical hH device, respectively.

Fig. S 1 The cooling properties of the (Bi,Sb)₂(Te,Se)₃-based and hH-based devices. a, Current dependent COP of (Bi,Sb)₂(Te,Se)₃-based device. **b,** Cooling heat flow (Q_c) of (Bi,Sb)₂(Te,Se)₃-based device. The BiTe materials are p-type Bi_{0.33}Sb_{1.67}Te₃ and n-type Bi₂Te_{2.7}Se_{0.3}. **c,** Current dependent COP and Q_c of (Bi,Sb)₂(Te,Se)₃-based device. **d,** COP dependent Q_c of hH-based device. The orange and olive lines indicate the high $Q_{c,max}$ with low COP and high COP with low Q_c conditions.

6. In page 8, the statement “Interestingly, κL did not increase with grain size due to the weak phonon scattering from the grain boundary” should be more clearly explained. As the grain boundary density is reduced with larger grains from the annealing, how is the κL lowered with weak phonon scattering?

Reply: Please kindly refer to the response of Question 3.

7. In the “High cooling capacity half-Heusler module performance” section, the leg lengths are ~ 0.2 mm and 0.3 mm (mm or cm). It is too small, and the authors are advised to check the same with the standard thermoelectric device leg length at Kelk and Thermion, respectively. As these values greatly influence the $Q_{c,max}$ calculations.

Reply: 0.2 mm is the lowest leg length used for advanced commercial devices. As shown in Fig. R3, the leg length of Kelk micro device used in optical communication unit is about 0.2 mm. Thermion also has developed micro-devices with leg-length of 0.2 and 0.3 mm (Fig. R4).

Fig. R3. Photograph of the Bi₂Te₃-based micro devices with leg length of ~0.2 mm from Kelk micro device.

Table 6. Parameters of new Thermion short-legged coolers

Model	No. of Couples	V_{\max} (V)	$Q_{c\max}$ (W)	Dimensions (mm)	
				Top	Bottom
1MC04-XXX-03 TE leg length 0.3mm					
Height - 1.7 mm (0.9 mm available), $I_{\max}=3$ A, $\Delta T_{\max}=69\pm 2$ K					
1MC04-004-03	4	0.5	0.75	1.4x2.8	2.8x2.8
1MC04-008-03	8	1	1.5	2.8x2.8	2.8x4.2
1MC04-012-03	12	1.5	2.3	2.8x4.2	4.2x4.2
1MC04-018-03	18	2.2	3.4	4.2x4.2	4.2x5.6
1MC04-032-03	32	3.9	6	5.6x5.6	5.6x7
1MC04-060-03	60	7.3	11	7x8.4	8.4x8.4
1MC04-XXX-02 TE leg length 0.2mm					
Height - 1.6 mm (0.8 mm available), $I_{\max}=4.5$ A, $\Delta T_{\max}=68\pm 2$ K					
1MC04-004-02	4	0.5	1.1	1.4x2.8	2.8x2.8
1MC04-008-02	8	1	2.2	2.8x2.8	2.8x4.2
1MC04-012-02	12	1.5	3.3	2.8x4.2	4.2x4.2
1MC04-018-02	18	2.2	5	4.2x4.2	4.2x5.6
1MC04-032-02	32	3.9	8.8	5.6x5.6	5.6x7
1MC04-060-02	60	7.3	16	7x8.4	8.4x8.4

Fig. R4. The Thermion micro-devices with leg-length of 0.2 and 0.3 mm (V. Semenyuk, "Thermoelectric micro modules for spot cooling of high density heat sources," *Proceedings ICT2001. 20 International Conference on Thermoelectrics*, Beijing, China, 2001, 391-396; V. Semenyuk, "Miniature Thermoelectric Modules with Increased Cooling Power," *2006 25th International Conference on Thermoelectrics*, Vienna, Austria, 2006, 322-326).

8. A suitable reference needs to be added for “the maximum cooling power density of the module ($Q_{c\max}$)” equation.

Reply: Thanks for the comment. We have added following reference accordingly.

In TE cooling, the maximum cooling heat flow density ($Q_{c_{max}}$) is governed by the relation³⁷:

$$Q_{c_{max}} \sim \frac{1}{2} PF \cdot T_c^2 - \Delta T \cdot \kappa \quad [1]$$

9. In-Page 5, the statement is included as “excellent conversion efficiency of 12% at ΔT of 600 K. The same is not reflected in Fig. 1 (c), as mentioned in the text. Authors must provide the necessary power efficiency estimation data in the supplementary information. Also, describe the contact resistance effect on the output power efficiency if you have considered it.

Reply: Thanks for the comment. All the discussion for power generation is presented in “High efficiency for power generation” section and data are presented in Fig. 6 and Fig. S11. Fig. 1c is only to illustrate the enhancement of power factor after defects and grain boundary engineering. No efficiency data is intended to be described in Fig. 1. We are sorry for the misunderstanding.

As the major target of this work is cooling application, we only used liquid metal as the contact layer and thermal interface material in this work, which provides excellent electrical and thermal contact. The details can be found in Supplementary Information — Materials and Methods section. It is quite challenging to measure the contact resistance using four probe system when the liquid metal is used. However, the measured resistance of uncouple module is generally consistent with predicted resistance using materials properties and dimensions of uncouple module, as shown in Fig. R5. As the electrical contact resistance is not considered in prediction, the agreement between prediction and experimental results indicates that the experimental electrical contact resistance is also negligible.

Fig. R5. Predicted and measured electrical properties of the hH uncouple device. **a**, The internal resistance. **b**, Open voltage. **c**, The deviation of internal resistance between prediction and measurement. The average deviation is $\sim 3.9\%$ which is indicated by the dot line.

We have modified the sentence to avoid any misunderstanding as below:

The minimized defects and grain boundary density enable substantial enhancement of mobility (Fig. 1a) and thus doubled PF up to $\sim 78 \mu\text{W cm}^{-1} \text{K}^{-2}$ near room temperature (Fig. 1c), which leads to the highest zT of ~ 0.4 at room temperature in hH alloys and highest average zT of 0.86 in the temperature range of 300 K to 873K among hH that is reflected in excellent power conversion efficiency of $\sim 12\%$ under ΔT of 600 K.

The agreement between prediction and experimental internal resistance (R_i) and open voltage (V_{oc}) of uni-couple device indicates that the experimental electrical contact resistance is negligible since the electrical contact resistance was not considered in prediction (Fig. 6b,c).

Fig. 6 Power generation performance of hH devices. **a**, Photo of the measurement setup for the high-performance hH unicycle module. **b-f**, Internal resistance (R_i) (b), open voltage (V_{oc}) (c), heat flow at zero current (Q_{cond}) (d), power density of TE legs (ω) (normalized to the length of 2 mm) (e), and conversion efficiency (η) (f) obtained using optimal current as a function of hot side temperature (T_h)^{34,38,51}. The radiation in the heat flow measurement is calibrated by the standard sample⁵⁰ (Fig. S11).

10. Fig. S3 (κ l) claims that the Sb-rich region from the grain boundary is eliminated. The segregated Sb is dissolved in the matrix after 2 days of annealing at 1143 K under the Sb pressure control. What is actually Sb pressure controlled? Does only Sb is regulated through an annealing process?

Reply: We introduced extra Nb foil with hot-pressed pellets in quartz tubes during the annealing process to promote the growth of grain. The Nb foil was used to react with evaporated Sb, and thereby the Sb pressure can be controlled and segregated Sb along grain boundaries is eliminated during the annealing. As shown in Fig. R6, the NbSb₂ phase was observed on the surface of Nb foil after Sb pressure-controlled annealing. While this effect cannot be achieved in conventional annealing process under vacuum condition. This is also why no significant grain growth was observed after vacuum annealing (Fig. S2). Besides Sb, we didn't observe any regulations of other elements during the

annealing process.

Fig. R6 XRD pattern of Nb foil before and after Sb pressure-controlled annealing. The XRD measurement was performed on the surface of Nb foil (*unpublished*).

11. In Fig.5 (d), the authors provided the power efficiency data for p-Nb_{0.55}Ta_{0.40}Ti_{0.05}FeSb and n-Zr_{0.44}Hf_{0.44}Ti_{0.12}NiSn_{0.9}Sb_{0.01} uni-couple, which is 40% higher than the previous work (Ref. 46). However, the contact resistance plays a vital role in the final device efficiency. Further, the authors did not provide any information about the uni-couple's contact engineering/contact resistance/total device resistance; the same need to be incorporated.

Reply: As the major target of this work is cooling application, we only used liquid metal as the contact layer and thermal interface material in this work, which provides excellent electrical and thermal contact. The details can be found in Supplementary Information — Materials and Methods section. It is quite challenging to measure the contact resistance using four probe system when the liquid metal is used. However, the measured resistance of uni-couple module is generally consistent with predicted resistance using materials properties and dimensions of uni-couple module, as shown in Fig. R5. As the electrical contact resistance is not considered in prediction, the agreement between prediction and experimental results indicates that the experimental electrical contact resistance is also negligible.

Fig. R5. Predicted and measured electrical properties of the hH uncouple device. **a**, The internal resistance. **b**, Open voltage. **c**, The deviation of internal resistance between prediction and measurement. The average deviation is $\sim 3.9\%$ which is indicated by the dot line.

We have modified Fig. 6 to provide internal resistance and open voltage of uni-couple device for contact resistance issue as below:

The agreement between prediction and experimental internal resistance (R_i) and open voltage (V_{oc}) of uni-couple device indicates that the experimental electrical contact resistance is negligible since the electrical contact resistance was not considered in prediction (Fig. 6b,c).

Fig. 6 Power generation performance of hH devices. **a**, Photo of the measurement setup for the high-performance hH uncouple module. **b-f**, Internal resistance (R_i) (b), open voltage (V_{oc}) (c), heat flow at zero current (Q_{cond}) (d), power density of TE legs (ω) (normalized to the length of 2 mm) (e), and conversion efficiency (η) (f) obtained using optimal current as a function of hot side temperature (T_h)^{34,38,51}. The radiation in the heat flow measurement is calibrated by the standard sample⁵⁰ (Fig. S11).

12. The COP formulae should be placed in the main/supplementary manuscript to explain the dependent and independent parameters for a better COP. Also, a detailed explanation needs to be added for the COP change concerning zT value and ΔT of the hH-YbAl3 device.

Reply: Thanks for the comment. As the current study is focusing on the high maximum cooling density under small ΔT for the thermal management of high-power electronics instead of high COP, we have mainly discussed the details about the Q_{cmax} in manuscript. As reviewer suggested, we have added the corresponding discussion in supplementary information to explain and address the concern of COP as below:

The COP value of the thermoelectric cooling device is determined by:

$$COP = \frac{ST_c I - K\Delta T - RI^2/2}{\Delta TI + RI^2} \quad (S.13)$$

where T_c , ΔT , I , S , K and R are the cold-side temperature, temperature difference, current, Seebeck coefficient, thermal conductance and the resistance of TE leg, respectively. Two most typical COPs to evaluate the performance of the TE cooling device are the COP of the maximum cooling density (COP_Q) and the COP under zero temperature difference (COP_0). The COP_Q decreases with increase of ΔT and z value, which is given by:

$$COP_Q = \frac{T_c}{2T_h} - \frac{\Delta T}{zT_hT_c} \quad (S.14)$$

Theoretically, when ΔT is zero, COP_Q is pinned at 0.5 which is obviously independent with thermoelectric properties of material. This why the material with high power factor demonstrated both high $Q_{c_{max}}$ and COP in device under low ΔT .

The COP_0 is given by:

$$COP_0 = \frac{ST_c}{RI} - \frac{1}{2} \quad (S.15)$$

It is obvious the electrical properties ($PF=S^2/\rho$) are important for the COP_0 , and the COP_0 decreases with increase of applied current. A more comprehensive discussion about COP can be referred to reference [6].

13. Authors should provide the contact resistance plots (Fig. 5(a)) for Nb_{0.55}Ta_{0.40}Ti_{0.05}FeSb and Zr_{0.44}Hf_{0.44}Ti_{0.12}NiSn_{0.9}Sb_{0.01} individual or device.

Reply: Please refer to Question 11. In addition, we have developed different brazing techniques and materials for half-Heusler-based thermoelectric modules that can be applied here (*Journal of Power Sources* 493 (2021) 229695; *ACS Appl. Mater. Interfaces* 2021, 13, 53935–53944). They all exhibit good contact and negligible contact resistance as shown in Fig. R7.

Fig. R7 Contact resistance of uni-couple hH device fabricated using different hH materials. a, $<1 \mu\Omega \cdot \text{cm}^2$ (*Journal of Power Sources* 493 (2021) 229695). b, $<3 \mu\Omega \cdot \text{cm}^2$ (*ACS Appl. Mater. Interfaces* 2021, 13, 53935–53944)

14. The authors claimed that “The maximum Q_{c_max} of $\sim 88 \text{ W cm}^{-2}$ can be achieved for hH-YbAl₃ device when the length of TE leg is projected to be 0.2 mm (Fig. 4f)” which is 210% higher than existing Bi₂Te₃ TE devices, under the identical conditions of $0.2 \mu\Omega/\text{cm}^2$, thermal conductance: $0.5 \text{ mm}^2\text{K/W}$, fill factor: 60% and these values calculated by changing the leg height. However, these are theoretical predictions, and obtaining such a low contact resistance is a big challenge in actual devices. Hence, the authors should mention a few probable candidate contact materials which can possess low specific contact resistance. Is there any data available with the authors for the said leg length (0.2 mm)? If yes, the same may be incorporated into the current manuscript.

Reply: Thanks for the comment. Metal/semiconductor contact has been well studied (*Materials Science Reports*, 1988, 3, 79). The contact properties are essentially determined by the width of the depletion region at the interface. High doping concentration in semiconductor will narrow the depletion layer and lead to the ohmic contact. Contact resistance value as low as $0.04 \mu\Omega\cdot\text{cm}^2$ can be achieved for Si-based contact system (Al/W/PtSi/p⁺-Si). Similar method to improve the contact has been adopted in thermoelectric device. By raising the surface carrier concentration, low contact resistivity of $0.24 \mu\Omega\cdot\text{cm}^2$ for p-type and $0.45 \mu\Omega\cdot\text{cm}^2$ for n-type materials in Bi₂Te₃ system can be obtained (*Advanced Thermoelectrics Materials, Contacts, Devices, and Systems*, 2017, CRC, 618).

Realizing low contact resistance for hH-YbAl₃ device is relatively easy due to the higher carrier concentration of hH and YbAl₃ as compared to Bi₂Te₃. A low contact resistance of $\sim 0.26 \mu\Omega\cdot\text{cm}^2$ for the junction of p-type Nb_{0.8}Ti_{0.2}FeSb and Mo electrode was reported by Tiejun Zhu et. al (*ACS Appl. Mater. Interfaces*, 2021, 13, 7317; *ACS Appl. Mater. Interfaces*, 2019, 11, 14182.). Furthermore, an idea metal/semiconductor contact can be formed with contact resistance less than $0.01 \mu\Omega\cdot\text{cm}^2$ (*APL Materials*, 2019, 7, 013202) by using full-Heusler as contact layer. YbAl₃ is a typical heavy-fermion metal with carrier concentration of $3\times 10^{22} \text{ cm}^{-3}$ (*J. Alloy Compd.*, 2017, 725, 1297). Therefore, an Ohmic contact can be possibly obtained as demonstrated in this work (Fig. S13.)

In current work, we only fabricate module with leg length of 7.7 mm. The devices with thin leg length, such as 0.2 mm, is currently under development.

15. The authors claimed that “excellent conversion efficiency of 12% at ΔT of 600 K for p-Nb_{0.55}Ta_{0.40}Ti_{0.05}FeSb and n-Zr_{0.44}Hf_{0.44}Ti_{0.12}NiSn_{0.9}Sb_{0.01} uni-couple”. The contact resistance will vary at the interface (metal contact-leg) with the temperature. Hence authors should provide the contact resistance experimental data for 600 K.

Reply: The four-point probe method has been widely applied to measure the contact resistance of thermoelectric devices (under room temperature). We agree with reviewer that the contact resistance may vary at elevated temperature. However, to our best knowledge, there is no such an instrument, either customized or commercial one, to measure the contact resistance at elevated temperature. All the previously reported data are measured at room temperature (e.g., *Nat. Commun.* 2021, 12.; *Energy* 2020, 191.; *Joule* 2020, 4, 2475.; *Energy Environ. Sci.* 2019, 12, 3390.; *Adv. Energy Mater.* 2022, 12, 2202392.; *Nat. Commun.* 2022, 13, 237. etc.). We would be very interested in such an instrument for high temperature contact resistance measurement if it is available.

The way to estimate the contact resistance at elevated temperature is to compare the measured resistance of device with predicted resistance based on the electrical resistivity and dimension of

materials. Please refer to the response of Question 9, 11 and 13.

REVIEWER COMMENTS

Reviewer #1 (Remarks to the Author):

Although the authors did their best to revise the manuscript and clarify my concerns, I did not feel this revision can be considered for publication in Nature Communications.

In No2 reply, the authors wrote two claims: both the maximum temperature difference under zero heat load and the maximum cooling density under zero temperature difference are the most important parameters to evaluate the performance of device. We focused on the application of thermal management for high-power electronics, which requires the maximum cooling density under small ΔT instead of ΔT maximization. But, if they want to emphasize “cooling density under zero temperature difference” and “maximum cooling density under small ΔT ”, the title should be modified. In general, for the thermoelectric research community and industrial engineers, ΔT maximization is commonly accepted.

Regarding my previous comment No5, the author added a figure (Fig. 3 in the revised ms) and a lot of words, but such a revision is really not satisfactory. My comment is “The authors spent a lot of spaces to emphasize the importance of grain growth, but it is also well known and the method has been reported“

Page 2, line 36: electrical thermal conductivity should be changed to electronic thermal conductivity

There are still many language mistakes:

Page 5, line 111: the hot-pressing temperature of Nb_{0.55}Ta_{0.40}Ti_{0.05}FeSb sample can only be conducted? at 1073 K,

Page 5, line 123: hot-pressed process

Page 6, line 139: As the grain growth are thermally activated

Reviewer #2 (Remarks to the Author):

The authors well-addressed my questions. Additionally, I also read the questions from other reviewers and the corresponding responses. It seems that the authors did a great job of improving the quality of the manuscript. This manuscript can now be accepted in its present form.

R#2's comment on R#1's report

As I have written in the previous comments, this manuscript presents a quite important and interesting work for thermoelectricity community because the mechanism given by the authors is important and the associated thermoelectric performance was validated by a national Lab. The authors also revised the manuscript by considering inputs from all the reviewers, which further makes this manuscript complete. As a result, it is surprising to me that reviewer #1 negatively evaluated the manuscript in this round of reviews.

I carefully read the input from the reviewer #1, and found that the reviewer #1 has two concerns about the work: if the maximum cooling density under small ΔT is important for thermoelectric cooling and the novelty of grain growth discussion in this study.

To me, the maximum temperature difference (ΔT maximization) and maximum cooling density under small ΔT represents two important aspects for thermoelectric cooling, based on specific application requirements. Previously, it is correct that ΔT maximization is commonly accepted in the thermoelectric research community as a materials performance (ZT) check. However, maximum cooling density under small ΔT becomes increasingly important since most of the thermoelectric applications require small ΔT and precise temperature control such electronics temperature control and in recent years, particularly demanded for hot-spots cooling. For instance, it is highly desired for laser diode cooling, which generates a significant amount of heat during operation and must be maintained at a specific temperature to ensure optimal performance and longevity. Currently available

Bi₂Te₃-based commercial modules are not able to provide enough cooling density and new technological directions (such as discussed in this manuscript) are required to advance the thermoelectric research in thermal management.

To reach to goal addressed above, the high power factor and high thermal conductivity of thermoelectric materials are required. The state-of-the-art Bi₂Te₃-based thermoelectric materials is still limited because of its low thermal conductivity and a moderate power factor. The half-Heusler material with high power factor and thermal conductivity that the authors have experimentally demonstrated in this study, thereby, is promising to address the challenge and demand on the thermal management of high-power electronic devices. It will potentially set new research directions in providing solutions to high power thermal management challenges. This is why I highly recommend Nature Communications to publish this manuscript.

I also remembered that the authors detailed the uniqueness of grain growth in this study, which reveals the pinning behavior of Sb-rich phase in grain boundary was eliminated through Sb pressure controlled annealing process. This has not been explored or observed before. It is also very rare for half-Heusler materials to achieve such significant grain growth. I think the content about the mechanism of grain growth discussed in revision is unique and in detail. Therefore, I regard it is one of novelties in this manuscript.

Considering the comments of reviewer #1, I would suggest that a further minor revision can be considered. In the revision, the authors are expected to address two things:

1. Make the application perspective, which is maximum cooling density under small ΔT , more clear. Why it is important and how it can be correlated to the materials proposed in this study need to be addressed. Accordingly, I agree with reviewer #1 that the title needs to be polished.
2. The language mistakes or typos need to be checked and corrected throughout the manuscript.

Reviewer #3 (Remarks to the Author):

The authors have incorporated the responses in their revised manuscript.

Dear Editor and Reviewers,

At the outset, we would like to thank you for providing us constructive feedback on our manuscript. The comments have been very helpful in further improving the quality of manuscript. We have revised our manuscript based on the reviewers' comments and questions. Hope you will find the changes to the manuscript satisfactory.

Below we provide point-by-point response to reviewer comments:

Reviewer #1 (Remarks to the Author):

Although the authors did their best to revise the manuscript and clarify my concerns, I did not feel this revision can be considered for publication in Nature Communications.

1. In No2 reply, the authors wrote two claims: both the maximum temperature difference under zero heat load and the maximum cooling density under zero temperature difference are the most important parameters to evaluate the performance of device. We focused on the application of thermal management for high-power electronics, which requires the maximum cooling density under small ΔT instead of ΔT maximization. But, if they want to emphasize “cooling density under zero temperature difference” and “maximum cooling density under small ΔT ”, the title should be modified. In general, for the thermoelectric research community and industrial engineers, ΔT maximization is commonly accepted.

Reply: Thank the reviewer for the suggestions. We have revised the title to “Can Half-Heusler Alloys be an Emerging High Power Density Thermoelectric Cooling Materials?” in this version. We appreciate your valuable input and hope that this revised title better reflects the focus of our work.

We agree that ΔT maximization (ΔT_{\max}) is a critical parameter that has been commonly accepted for thermoelectric devices as it is directly related to the zT value of the thermoelectric materials. **However, for the hot spot cooling (e.g. on-chip hot spot cooling), cooling density is also crucial because zT is not the relevant parameter and higher zT values do not result in pumping more heat.** In fact, **ΔT_{\max} and Q_{cmax} ($\Delta T = 0$) are both used to evaluate the performance of TE cooling devices in the industrial community** (Figures R1 and Figure R2). In this case, we believe the Q_{cmax} deserves more attention in research community. It is well known that high Q_{c} thermoelectric devices are crucial for high-power electronics, such as optoelectronics and hotspot thermal management, where the typical working condition is under small ΔT . Indeed, we need to combine active and passive cooling and design a thermoelectric material with large power factor as well as large thermal conductivity for high-power electronic cooling (*Appl. Phys. Lett.* 2015, **106**, 203506). The cooling performance of the hH-YbAl₃ and Bi₂Te₃-based devices are given and compared in Figure R3. Clearly, the Bi₂Te₃-based device shows higher performance under high ΔT due to the higher zT near room temperature. However, hH-YbAl₃ device shows outstanding advantages when working with high Q_{c} and low ΔT . This underscores the importance of this work.

A Nominal Performance in Nitrogen

Hot Side Temperature (°C)	27	85
ΔT_{max} (°C):	71	90
Q_{max} (watts):	1.5	1.8
I_{max} (amps):	0.9	0.9
V_{max} (vdc):	2.4	3.0
AC Resistance (ohms):	2.15	--

(Th=50°C)

TEM Model No.	I_{max} (A)	V_{max} (V)	ΔT_{max} (°C)	Q_{max} (W)	SIZE(mm)			
					W	L1	L2	H
72013/127/030B	3.0	18.1	83	28	29.7	29.7		3.80
72013/127/040B	4.0	18.1	83	38	29.7	29.7		3.80
7200A/127/040B	4.0	18.1	83	38	39.7	39.7		4.50
72005/071/060B	6.0	10.1	83	32	29.8	29.8		4.00
72005/127/060B	6.0	18.1	83	57	39.7	39.7		4.00
72005/128/060B	6.0	18.2	83	57	39.7	39.7	42.8	4.00
72003/071/085B	8.5	10.1	83	45	29.8	29.8		3.80
72003/127/085B	8.5	18.1	83	80	39.7	39.7		3.80
7200A/031/090B	9.0	4.4	83	21	29.8	29.8		4.50
7201A/032/100B	10.0	4.5	83	24	25.4	25.4	28.7	4.50
7200A/031/150B	15.0	4.4	83	35	29.8	29.8		4.50
72013/032/150B	15.0	4.5	83	36	25.4	25.4	28.7	3.80
72058/199/160B	16.0	28.3	83	237	40.0	58.0		3.30

C Model KSGH018 RoHS 2002/95/EC Compliant

Th	ΔT_{max} (°C)	I_{max} (A)	V_{max} (V)	Q_{max} (W)
27°C	76	1.5	2.2	1.8
70°C	93	1.5	2.8	2.0

	W	L1	L2	H
Size(mm)	2.00	2.00	2.60	0.70
Tolerance(mm)	±0.2	±0.2	±0.2	±0.1
Metallization	Cu-Ni-Pd-Au			
Ceramic material	Aluminum nitride (AlN)			
Assembly solder	AuSn (melting point:280°C)			

Model	H (mm)	I_{max} (A)	ΔT_{max} (K)	Q_{max} (W)
1TMC04-007-XX				
$N=7$ $Top = Base = AxB = 3.2 \times 3.2$ $U_{max}=0.84 V$				
1TMC04-007-15	2.65	0.55	74	0.27
1TMC04-007-12	2.35	0.7	74	0.33
1TMC04-007-10	2.15	0.8	74	0.39
1TMC04-007-08	1.95	1	74	0.48
1TMC04-007-05	1.65	1.55	73	0.72

Figure R1. The parameters used to evaluate the performance of TE cooling devices in different companies. A) Malow, B) Ferrotec, C) Kelk, D) Thermion.

Understanding TE Cooler General Parameters, dT_{max} and Q_{max}

dT_{max} for TEC is specified without Heatload. Q_{max} is specified at $dT=0$. The application point is in between.

Copyright TEC Microsystems GmbH. Images contain hidden watermark.

Figure R2. Sample performance chart, provided by TEC Microsystems company.

Figure R3. Performance chart of hH-YbAl₃ and Bi₂Te₃ devices reported in our work.

- Regarding my previous comment No5, the author added a figure (Fig. 3 in the revised ms) and a lot of words, but such a revision is really not satisfactory. My comment is “The authors spent a lot of spaces to emphasize the importance of grain growth, but it is also well known and the method has been reported.

Reply: The study of grain boundary scattering in hH system mainly focuses on the scattering mechanism (*J. Appl. Phys.* 2013, **114**, 134905 and *Energy Environ. Sci.*, 2022, **15**, 1406-1422). However, **achieving improved mobility with promoted thermoelectric performance by enlarged grain size in main hH families is still a challenging** due to the high melting point of hH and the alloy elements, which make grain boundary migration difficult. In this study, we focused on the (Nb,Ta)FeSb hH alloys and successfully achieved the highest zT_{avg} with significant PF improvement through a Sb pressure-controlled annealing process. **This method is new and results in a three-order-of-magnitude increase in grain size (as reviewer #2 commented).** Moreover, the increase power factor and thermal conductivity, as shown in Figure 4 b and d, meet the criterion of TE cooling device toward high-power electronics (*Appl. Phys. Lett.* 2015, **106**, 203506).

The Sb pressure-controlled annealing process developed in this study has a significant effect on promoting grain growth and can potentially be extended to crystals growth of other alloys and compounds. In our previous work, we calculated that (Nb,Ta)FeSb is a typical material with weak electron-phonon interactions (*Nat Commun* 2018, **9**, 1721), indicating its promising potential for room temperature cooling applications. While we achieved facile grain growth of lightly doped NbFeSb (*Adv. Sci.* 2018, **5**, 1800278 and *Proc. Natl. Acad. Sci. U.S.A.* 2016, **113**, 13576-13581), its inferior zT inhibit its application at room temperature. The high zT value of NbFeSb requires high-concentration alloying or doping, both of which make the strategy of reducing boundary scattering difficult to be realized. To achieve high room temperature zT and PF and to verify the cooling ability of hH, we devoted considerable efforts over the years to obtain

the coarse-grain (Nb,Ta)FeSb until the pinning behavior of the Sb-rich phase at the boundary was finally revealed. We hope the aforementioned information could explain the importance of the dynamic mechanism of grain growth and synthesis method discovered and demonstrated in this work.

3. Page 2, line 36: electrical thermal conductivity should be changed to electronic thermal conductivity

There are still many language mistakes:

Page 5, line 111: the hot-pressing temperature of Nb_{0.55}Ta_{0.40}Ti_{0.05}FeSb sample can only be conducted? at 1073 K,

Page 5, line 123: hot-pressed process

Page 6, line 139: As the grain growth are thermally activated

Reply: We have carefully gone through the entire manuscript and corrected all the typos and minor mistakes.

Reviewer #2 (Remarks to the Author):

The authors well-addressed my questions. Additionally, I also read the questions from other reviewers and the corresponding responses. It seems that the authors did a great job of improving the quality of the manuscript. This manuscript can now be accepted in its present form.

R#2's comment on R#1's report

As I have written in the previous comments, this manuscript presents a quite important and interesting work for thermoelectricity community because the mechanism given by the authors is important and the associated thermoelectric performance was validated by a national Lab. The authors also revised the manuscript by considering inputs from all the reviewers, which further makes this manuscript complete. As a result, it is surprising to me that reviewer #1 negatively evaluated the manuscript in this round of reviews.

I carefully read the input from the reviewer #1, and found that the reviewer #1 has two concerns about the work: if the maximum cooling density under small ΔT is important for thermoelectric cooling and the novelty of grain growth discussion in this study.

To me, the maximum temperature difference (ΔT maximization) and maximum cooling density under small ΔT represents two important aspects for thermoelectric cooling, based on specific application requirements. Previously, it is correct that ΔT maximization is commonly accepted in the thermoelectric research community as a materials performance (ZT) check. However, maximum cooling density under small ΔT becomes increasingly important since most of the thermoelectric applications require small ΔT and precise temperature control such electronics temperature control and in recent years, particularly demanded for hot-spots cooling. For instance, it is highly desired for laser diode cooling, which generates a significant amount of heat during operation and must be maintained at a specific temperature to ensure optimal performance and longevity. Currently available Bi₂Te₃-based commercial modules are not able to provide enough cooling density and new technological directions (such as discussed in this manuscript) are required to advance the thermoelectric research in thermal management.

To reach to goal addressed above, the high power factor and high thermal conductivity of thermoelectric materials are required. The state-of-the-art Bi₂Te₃-based thermoelectric materials is still limited because of its low thermal conductivity and a moderate power factor. The half-Heusler material with high power factor and thermal conductivity that the authors have experimentally demonstrated in this study, thereby, is promising to address the challenge and demand on the thermal management of high-power electronic devices. It will potentially set new research directions in providing solutions to high power thermal management challenges. This is why I highly recommend Nature Communications to publish this manuscript.

I also remembered that the authors detailed the uniqueness of grain growth in this study, which reveals the pinning behavior of Sb-rich phase in grain boundary was eliminated through Sb pressure controlled annealing process. This has not been explored or

observed before. It is also very rare for half-Heusler materials to achieve such significant grain growth. I think the content about the mechanism of grain growth discussed in revision is unique and in detail. Therefore, I regard it is one of novelties in this manuscript.

Considering the comments of reviewer #1, I would suggest that a further minor revision can be considered. In the revision, the authors are expected to address two things:

1. Make the application perspective, which is maximum cooling density under small ΔT , more clear. Why it is important and how it can be correlated to the materials proposed in this study need to be addressed. Accordingly, I agree with reviewer #1 that the title needs to be polished.
2. The language mistakes or typos need to be checked and corrected throughout the manuscript.

Reply: We appreciate the suggestions and support from the reviewer. We have revised the title to “Can Half-Heusler Alloys be an Emerging High Power Density Thermoelectric Cooling Materials?” in this version. The significance of high cooling density applications under small ΔT and the requisite thermoelectric properties has been discussed in this version. And we have carefully gone through the entire manuscript and corrected all the typos and minor mistakes.

REVIEWERS' COMMENTS

Reviewer #1 (Remarks to the Author):

The authors treated my comments seriously and revised the manuscript accordingly. This version should be better. But I still feel the following points should be considered.

The first 7 lines of the abstract should be reconsidered. In particular, the sentence "However, current state-of-the-art Bi₂Te₃-based TE materials show limited cooling power density" should be removed. I do not think Figure 3 is needed in the text, may be okay as supporting materials. Caption of Figure 2: Growth of grain size?

Dear Editor and Reviewers,

At the outset, we would like to thank you for providing us constructive feedback on our manuscript. The comments have been very helpful in further improving the quality of manuscript. We have revised our manuscript based on the reviewers' comments. Below we provide point-by-point response:

Reviewer #1 (Remarks to the Author):

The authors treated my comments seriously and revised the manuscript accordingly. This version should be better. But I still feel the following points should be considered. The first 7 lines of the abstract should be reconsidered. In particular, the sentence "However, current state-of-the-art Bi₂Te₃-based TE materials show limited cooling power density" should be removed. I do not think Figure 3 is needed in the text, may be okay as supporting materials. Caption of Figure 2: Growth of grain size?

Reply: Thank you for your valuable feedback and support, which we greatly appreciate. We have taken your suggestions into consideration and made the necessary revisions to improve the quality of our manuscript. Specifically, we have removed Figure 3 and corresponding discussion to the supporting materials and corrected the typos and minor mistakes. Additionally, we have revised the abstract accordingly as below.

To achieve optimal thermoelectric performance, it is crucial to manipulate the scattering processes within materials to decouple the transport of phonons and electrons. In half-Heusler (hH) compounds, selective defect reduction can significantly improve performance due to the weak electron-acoustic phonon interaction. This study utilized Sb-pressure controlled annealing process to modulate the microstructure and point defects of Nb_{0.55}Ta_{0.40}Ti_{0.05}FeSb compound, resulting in a 100% increase in carrier mobility and a maximum power factor of 78 $\mu\text{W cm}^{-1} \text{K}^{-2}$, approaching the theoretical prediction for NbFeSb single crystal. This approach yielded the highest average zT of ~0.86 among hH in the temperature range of 300-873 K. The use of this material led to a 210% enhancement in cooling power density compared to Bi₂Te₃-based devices and a conversion efficiency of 12%. These results demonstrate a promising strategy for optimizing hH materials for near-room-temperature thermoelectric applications.